# Sex and species specific hearing mechanisms in mosquito flagellar ears

Matthew P. Su [1,2,3], Marta Andrés[1,3], Nicholas Boyd-Gibbins [1,4], Jason Somers [1,3] & Joerg T. Albert [1,2,3,5]

Hearing is essential for the courtship of one of the major carriers of human disease, the mosquito. Males locate females through flight-tone recognition and both sexes engage in mid-air acoustic communications, which can take place within swarms containing thousands of individuals. Despite the importance of hearing for mosquitoes, its mechanisms are still largely unclear. We here report a multilevel analysis of auditory function across three disease-transmitting mosquitoes (*Aedes aegypti*, *Anopheles gambiae* and *Culex quinquefasciatus*). All ears tested display transduction-dependent power gain. Quantitative analyses of mechanotransducer function reveal sex-specific and species-specific variations, including male-specific, highly sensitive transducer populations. Systemic blocks of neurotransmission result in large-amplitude oscillations only in male flagellar receivers, indicating sexually dimorphic auditory gain control mechanisms. Our findings identify modifications of auditory function as a key feature in mosquito evolution. We propose that intra-swarm communication has been a driving force behind the observed sex-specific and species-specific diversity.

[1] Ear Institute, University College London, 332 Gray's Inn Road, London WC1X 8EE, UK. [2] Centre for Mathematics and Physics in the Life Sciences and Experimental Biology (CoMPLEX), University College London, Gower Street, London WC1E 6BT, UK. [3] The Francis Crick Institute, 1 Midland Road, London NW1 1AT, UK. [4] Present address: Center for iPS Cell Research and Application, Kyoto University, 53 Kawahara-cho, Shogoin, Sakyo-ku, Kyoto 606-8507, Japan. [5] Present address: Department of Cell and Developmental Biology, University College London, Gower Street, London WC1E 6DE, UK. Correspondence and requests for materials should be addressed to J.T.A. (email: joerg.albert@ucl.ac.uk)

Mosquito-borne diseases are responsible for hundreds of thousands of deaths every year, with significant associated morbidities[1]. Whilst mosquito control programmes have successfully reduced disease-associated mortality and morbidity since 2000, they are now facing increasing pressure from (amongst other factors) insecticidal resistance[2]. New control strategies are needed and targeting mosquito reproductive behaviour has been identified as a promising, yet underexploited, possibility[3]. Hearing plays a key role in mosquito courtship[4]; a deeper knowledge of its mechanistic bases is thus not only a prerequisite for understanding mosquito acoustic communication but could also help the development of novel control tools.

Mosquito flagellar ears are comprised of two functionally distinct components: (i) the flagellum, which forms an inverted pendulum and acts as the sound receiver and (ii) the Johnston's organ (JO), a chordotonal organ (ChO)[5,6], which is the site of auditory transduction. JO neurons are ciliated mechanosensory cells that are connected to prongs at the base of the flagellum. These neurons are stretch-activated by deflections of the flagellar sound receiver (see Fig. 1a). With >15,000 neurons, the JOs of male mosquitoes are the largest ChOs reported in insects[7]; the JOs of female mosquitoes contain around half this number[8]. Therefore, both the neuroanatomy[7,8] and reported response sensitivity of the female ear[9], as well as the existence of intersexual acoustic communication[10–13], suggest that hearing plays vital roles in both males and females.

Males of many mosquito species form swarms of varying sizes that females then enter in order to copulate[14–16]. In terms of acoustic communication between the sexes, mosquito swarms are highly asymmetric environments: tens, hundreds, or (in the case of *Anopheles gambiae*) sometimes thousands of males listen out for the flight tone of individual females entering the swarm[15]. Swarms thus form part of the mosquitoes' natural acoustic space and their corresponding signal-to-noise ratios, as well as resulting amplification and filtering challenges, can be expected to be vastly different for male and female ears. Several studies have proposed potential mechanisms of acoustic signalling between conspecific males and females[10–13,17,18], but few have discussed these within the context of flying animals[19,20] or related these to the specific environment of the swarm[19]. Current reports hypothesise that males detect and locate conspecific females by listening out for the female's flight tones and dynamic interactions between male and female flight tones mediate pre-copulatory interactions[3].

In both vertebrates and insects, ears have evolved as active sensors in response to the sensory ecological needs of their environments[21,22]. Reflecting the specific mode of operation of all ears, that is, direct activation by sound-associated forces, large parts of the filtering, amplification, and processing of sound already happen at the level of the auditory cells (namely the auditory transducer ion channels that open and close in response to sound). We therefore tested if the asymmetric acoustic environment of mosquito swarms is reflected in sexually dimorphic transduction mechanisms and/or variations of the previously reported efferent innervation of the mosquito ear[23].

Another phenomenon that might offer valuable insights into mosquito auditory function (and indeed acoustic courtship) are spontaneously occurring, self-sustained oscillations (SOs) of the flagellum. SOs are large (~1000 times above baseline), almost mono-frequent flagellar oscillations that persist independent of external sound stimulation and seem to be restricted to males[9]. While mosquito SOs have been induced by non-specific physiological impairments, for example, dimethyl sulfoxide injection[9], no physiologically specific induction of SOs has yet been reported. It has therefore remained unclear whether SOs in mosquitoes reflect a pathological signature or a key mechanism of active hearing. SOs could, for example, aid males in the localisation of

conspecific females by boosting the ear's sensitivity to the frequency of the female wingbeat, thus amplifying the faint sound emissions of flying females[17].

In order to better understand the connections between mosquito auditory behaviour and the molecular and biophysical operation of their flagellar ears, we investigated auditory function in three major mosquito vectors of human disease: the two Culicine species, *Aedes aegypti* (vector of dengue and Zika virus) and *Culex quinquefasciatus* (West Nile virus, *Wuchereria bancrofti*), and the Anopheline species, *Anopheles gambiae* (malaria).

The ears of all mosquitoes tested exhibit power gain, that is, they actively inject energy into mechanically evoked receiver vibrations. Similar to hearing in vertebrates[24] and fruit flies[25], mosquito hearing relies on directly gated mechanotransducer modules. In-depth quantitative analyses reveal substantial degrees of sex-specific and species-specific variation, including male-specific populations of highly sensitive transducers. Compounds known to ablate ChO mechanotransduction[26,27] eliminate both auditory energy injection and mechanical signatures of transducer gating in mosquitoes. Blocking systemic neurotransmission results in large SOs only in male antennae, increasing their power gain by more than three orders of magnitude. We also find that SOs entrain (i.e. they adopt the oscillation frequency of an external stimulus) only to pure tones close to female wingbeat frequencies. We suggest that SOs in male flagellar ears play a key role in the extraction and amplification of female wingbeat signals and that mosquito auditory systems are viable targets for vector control programmes.

## Results

**A transduction-dependent amplifier supports mosquito hearing.** We first analysed the vibrations of unstimulated mosquito sound receivers (free fluctuations); these have previously been used to assess frequency tuning and amplification in the fly's auditory system[28,29].

Using a modified version of the framework provided by Göpfert et al.[28], we compared the total flagellar fluctuation powers of metabolically challenged ($CO_2$-sedated/$O_2$-deprived or passive) animals to those of metabolically enabled ($O_2$-supplied or active) ones. In both sexes of all three species, flagellar fluctuation powers were significantly higher in the active, metabolically enabled state (Fig. 1b; Supplementary Figure 1a, b), demonstrating power gain, that is, active injection of energy, for the mosquito flagellar ear (Figure 1c and Table 1).

Baseline energy injections (defined as energy content above thermal energy; in $k_BT$) were significantly different between males and females only for *Cx. quinquefasciatus* (analysis of variance (ANOVA) on ranks, $p < 0.05$). Median values for *Cx. quinquefasciatus* males were estimated at 1.85 (SEM: $\pm 2.40$)$k_BT$ ($N = 31$) compared to 6.26 (SEM: $\pm 2.05$)$k_BT$ for conspecific females ($N = 28$). Furthermore, *Cx. quinquefasciatus* females injected significantly more energy than any other species or sex tested (ANOVA on ranks, $p < 0.01$ in all cases; Table 1); no other significant differences were identified (ANOVA on ranks, $p > 0.05$ in all cases).

Free fluctuation recordings also allow for extraction of two other key parameters of auditory function in both active and passive states (Table 1): the best frequency, $f_0$, and the tuning sharpness, $Q$, of the flagellum.

Flagellar best frequencies were not significantly different between active and passive states for female *Cx. quinquefasciatus* or *Ae. aegypti*; the flagellar best frequency for female *An. gambiae* was, however, lower in the active state than in the passive state (ANOVA on ranks, $p < 0.001$). In contrast to this, the flagellar best frequency of males from all three species was higher in the active state than the passive (ANOVA on ranks, $p < 0.001$).

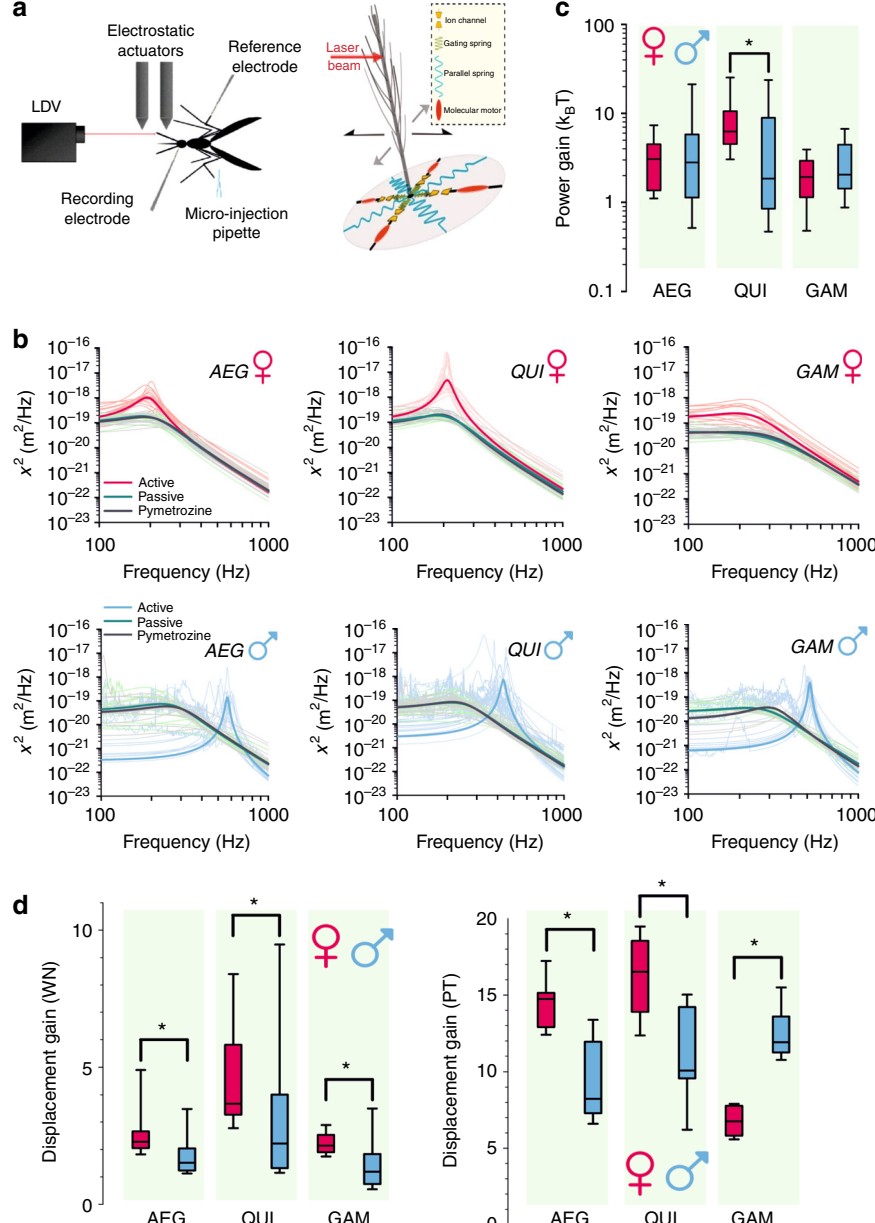

**Fig. 1** Transducer-based amplification in mosquito ears. **a** Experimental paradigm of laser Doppler vibrometry (LDV) recordings (left) and transducer sketch of mosquito flagellum (right), with the laser beam focussed on the flagellum—black arrows represent movement in the plane of the laser beam, grey arrows represent potential flagellar motion in other planes. In-figure legend describes individual components of sketch (adapted from ref. [22]). **b** Power spectral densities (PSDs) from harmonic oscillator fits to free fluctuations of female and male flagella (*Ae. aegypti* (AEG), *Cx. quinquefasciatus* (QUI), and *An. gambiae* (GAM)) in three separate states: active, passive and pymetrozine exposed. Prominent solid lines represent fits created from median parameter values (i.e. median values for a specific group), while shaded lines represent damped harmonic oscillator fits for individual mosquitoes. **c** Box-and-whisker plots for calculated power gains for flagellar receivers of females and males— significant differences (ANOVA on ranks, $p < 0.05$) between conspecific female and male mosquitoes are starred. Centre line, median; box limits, lower and upper quartiles; whiskers, 5th and 95th percentiles. Sample sizes: *Ae. aegypti* females = 35; *Ae. aegypti* males = 29; *Cx. quinquefasciatus* females = 28; *Cx. quinquefasciatus* males = 31; *An. gambiae* females = 33; *An. gambiae* males = 24. **d** Displacement gain values estimated using white noise (WN, intensity-dependent displacement gain, top) or pure tone (PT, frequency-dependent displacement gain, bottom) stimulation for female and male *Ae. aegypti* (AEG), *Cx. quinquefasciatus* (QUI) and *An. gambiae* (GAM), with significant differences between conspecific females and males starred (Mann–Whitney rank-sum tests, $p < 0.05$). Centre line, median; box limits, lower and upper quartiles; whiskers, 5th and 95th percentiles. Sample sizes (WN/PT): *Ae. aegypti* females = 7/8; *Ae. aegypti* males = 7/10; *Cx. quinquefasciatus* females = 13/8; *Cx. quinquefasciatus* males = 13/8; *An. gambiae* females = 9/7; *An. gambiae* males = 7/7

For all species investigated, the frequency tuning was significantly sharper (and corresponding Q values higher) in males than in females; flagellar tuning was also sharper in active as compared to the passive states (Table 1).

Administration of pymetrozine, an insecticide known to specifically eliminate ChO mechanotransduction[26,27], resulted in the flagellar receivers of all mosquitoes tested becoming similar to their passive states; power gain was abolished (Fig. 1b and

**Table 1 Principal parameters from free fluctuation analysis**

| | AEG ♀ | AEG ♂ | QUI ♀ | QUI ♂ | GAM ♀ | GAM ♂ |
|---|---|---|---|---|---|---|
| **Active state** | | | | | | |
| Sample size | 52 | 39 | 37 | 43 | 42 | 35 |
| Best frequency (Hz) | 203.06 (2.22) | 522.69 (11.10) | 212.96 (2.41) | 485.40 (7.03) | 219.70 (3.55) | 506.62 (9.03) |
| Tuning sharpness $Q$ | 3.32 (0.21) | 26.80 (1.57) | 6.17 (3.17) | 17.77 (1.62) | 1.19 (0.24) | 21.59 (2.51) |
| **Passive state** | | | | | | |
| Sample size | 35 | 29 | 28 | 35 | 33 | 24 |
| Best frequency (Hz) | 207.47 (3.99) | 297.99*** (11.90) | 206.45 (3.32) | 273.76*** (10.23) | 325.00*** (7.44) | 283.68*** (5.69) |
| Tuning sharpness $Q$ | 1.04*** (0.04) | 0.94*** (0.06) | 1.11*** (0.04) | 1.00*** (0.05) | 0.67*** (0.03) | 0.91*** (0.07) |
| **Pymetrozine state** | | | | | | |
| Sample size | 30 | 25 | 27 | 35 | 26 | 18 |
| Best frequency (Hz) | 210.21 (3.57) | 292.67*** (11.49) | 208.93 (1.90) | 258.68*** (8.65) | 305.99*** (7.07) | 319.12*** (10.65) |
| Tuning sharpness $Q$ | 1.31*** (0.05) | 1.27***,† (0.08) | 1.59***,†† (0.10) | 1.16*** (0.05) | 0.94**,† (0.04) | 1.66*** (0.30) |
| Apparent flagellar mass (ng) | 40.54 (2.59) | 43.93 (3.00) | 32.32 (1.44) | 39.18 (2.20) | 45.35 (2.64) | 54.14 (3.46) |
| Power gain ($k_BT$) | 3.06 (0.62) | 2.81 (1.69) | 6.26 (2.05) | 1.85 (2.40) | 1.93 (0.25) | 2.05 (0.58) |

Median values obtained from free fluctuation fits of female and male *Ae. aegypti* (AEG), *Cx. quinquefasciatus* (QUI) and *An. gambiae* (GAM) flagella (standard errors in brackets); values include flagellar best frequency, tuning sharpness ($Q$), apparent mass and estimated power gain in the quiescent state (i.e. not displaying SOs). Significant differences between the active state and any other state (passive or pymetrozine exposed) for a specific mosquito group are starred (ANOVA on ranks; \*\**p* < 0.01; \*\*\**p* < 0.001). Significant differences between the passive state and pymetrozine-exposed state for a specific mosquito group are also highlighted (ANOVA on ranks; †*p* < 0.05; ††*p* < 0.01). Recordings were made at 22 °C; further experimental conditions are detailed in the Methods section

Table 1). Flagellar best frequency and tuning sharpness were also similar to those observed in the passive state.

The preceding experiments extracted baseline properties of the mosquito ear from unstimulated flagellar receivers only. We therefore extended our analyses to cover a wider range of auditory function using two stimulus types: different intensities of white noise (upper limit 3200 Hz) and different frequencies of pure tones (15–695 Hz). Such comparative stimulus–response analyses can produce insights of immediate ecological relevance; this is particularly valid for pure tones, which closely mimic the sounds emitted by flying mosquitoes.

Concretely, the two stimulus types allowed for the calculation, and comparison, of the receivers' intensity-dependent (for white noise) and frequency-dependent (for pure tones) displacement gains (Fig. 1d). These dimensionless displacement gains are calculated as the fold-difference in flagellar displacement sensitivities (measured as a ratio of displacement over force) between the respective sensitivity maxima and minima. For broadband, white noise stimulation, the value thus describes how much higher the sensitivity is for the smallest as compared to the largest stimuli, reflecting the characteristic intensity dependence of transducer-based auditory amplification[30] (Fig. 1d, top; Supplementary Figure 1c, top). For narrowband, pure tone stimulation (at mid-range intensity), the values describe how much higher the sensitivity is at the flagellar resonance as compared to off-resonance frequencies (Fig. 1d, top; Supplementary Figure 1c, bottom).

Significant differences were observed in the receivers' displacement gains: (i) in all species, females display significantly higher displacement gains than their male counterparts for white noise stimulation (Fig. 1d, top) (Mann–Whitney rank-sum tests, *p* < 0.05); (ii) for pure tone stimulation, culicine females displayed significantly higher displacement gains than conspecific males, whereas the situation was reversed in the anopheline species *An. gambiae* (Fig. 1d, bottom) (Mann–Whitney rank-sum tests, *p* < 0.05).

White noise stimulation also allowed for quantifying previously observed, intensity-dependent changes of flagellar best frequencies (Supplementary Figure 1d). The flagellar best frequencies of both culicine females showed only small (<10%) intensity-dependent modulations with no clear signs of an intensity-dependent increase or decrease. The receivers of *An. gambiae* females, however, showed characteristic intensity-dependent best

frequency increases as previously reported for *Drosophila*[30,31]. Male flagellar best frequencies, in contrast, remained constant up to a distinct force intensity, and then decreased to a new level.

Taken together, these analyses reveal substantial degrees of sex-specific and species-specific variation in response to different types of auditory stimuli.

**Sex-specific and species-specific transduction in mosquito ears.** In order to probe mosquito auditory transduction directly we again adapted a paradigm previously devised for *Drosophila*[25]. Force steps electrostatically applied to mosquito flagellar receivers were used to quantify mechanical signatures of auditory transducer gating. In parallel to these mechanical analyses, we also recorded mechanically evoked compound action potential (CAP) responses from the mosquitoes' antennal nerves (Supplementary Figure 2a contains examples of flagellar and auditory nerve responses to force steps).

An essential consequence of direct, mechanical transducer gating is that the receiver structures coupled to the transducers will display *gating compliances*, that is, they will be more compliant (or less stiff) over the range of forces and displacements where transducer gating occurs[24]. The various nonlinearities reported for mosquito flagellar receivers are consistent with the existence of functionally relevant gating compliances[32], but auditory transducer mechanics has not been probed directly in mosquitoes before.

We quantified flagellar stiffness by calculating the partial differential of force with respect to displacement in response to force-step actuation. The flagellar receivers of female mosquitoes from all three species showed distinct decreases in stiffness, that is, increases in compliance, around the resting position in a similar (if lesser) manner to *Drosophila*[25] (Fig. 2a). The largest changes in flagellar stiffness were found for *An. gambiae* females (Fig. 2a, bottom left), which also show a significant shift in flagellar best frequency between active and passive states (Table 1); such shifts have been reported as another signature of direct transducer gating[30].

Nerve response curves closely followed the flagellar compliance patterns (Fig. 2b) with recorded CAP magnitudes well matching mechanically predicted transducer channel open probabilities (Fig. 2b), once again in good agreement with previous reports from *Drosophila*[25]. Female *An. gambiae* produced significantly

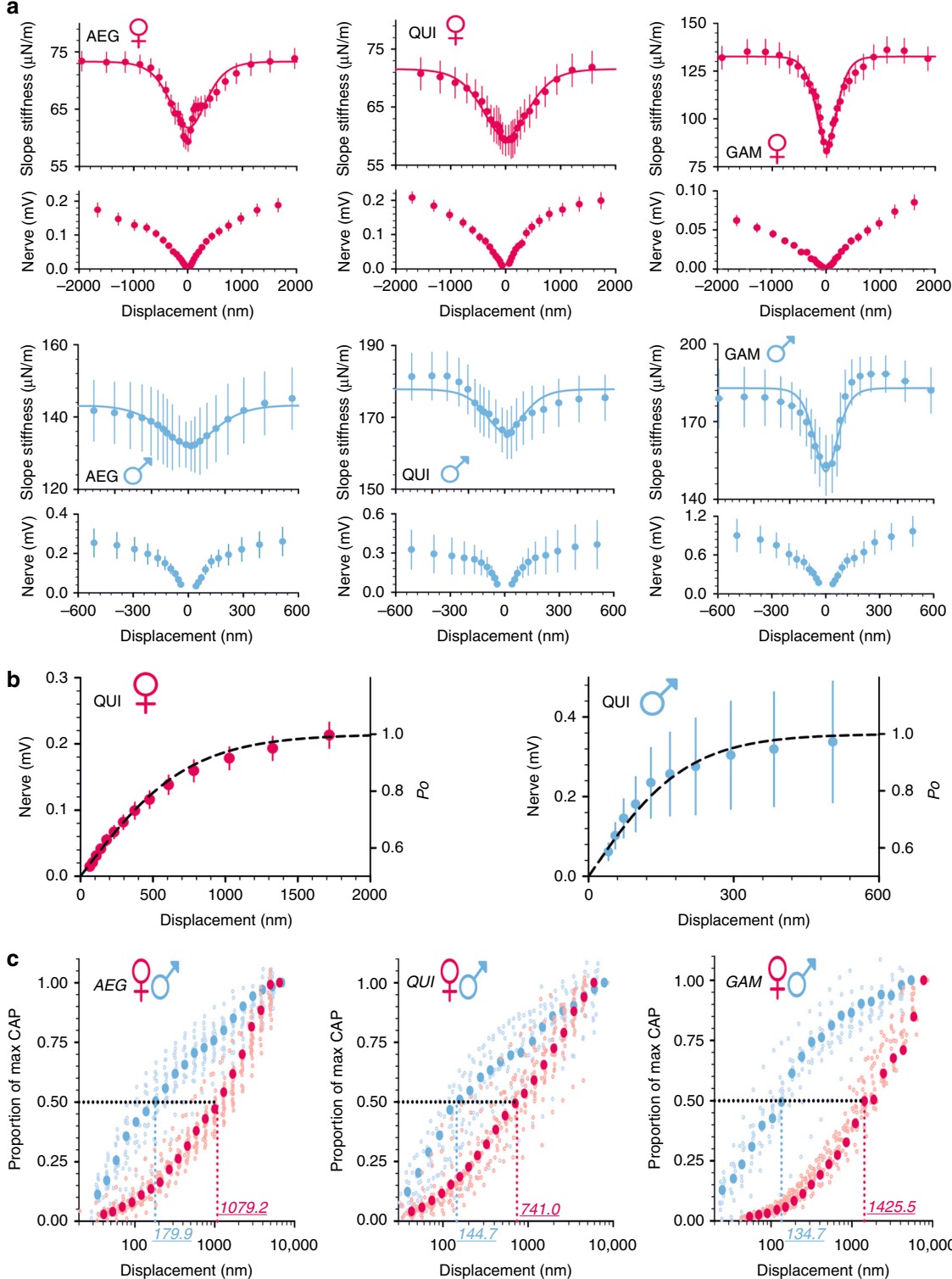

smaller magnitude CAP responses than females from the two other species (ANOVA on ranks, $p < 0.001$ in all cases; Fig. 2b).

Around their resting positions, the flagellar receivers of males (Fig. 2a, right) also showed characteristic nonlinear compliances (or decreases in stiffness), which aligned well with a first saturating nonlinearity in the corresponding CAP responses. Compared to their conspecific females, however, male mosquitoes across all species had significantly higher values for all relevant stiffness parameters (ANOVA on ranks, $p < 0.05$); these include the flagellar steady-state stiffness, $K_{STEADY}$, the asymptotic

stiffness, $K_{INFINITY}$, and the total gating spring stiffness, $K_{GS}$ (Table 2 and Supplementary Figure 2b)[33].

Moreover, CAP magnitudes were substantially larger for all males than for conspecific females at equivalent displacements or forces (Fig. 2a, c). This is particularly evident for displacements $\lesssim 250$ nm (at measurement point), where male CAP responses are almost one order of magnitude larger.

Following pymetrozine exposure, all mechanical signatures of transducer gating were abolished. Flagellar stiffness values for female *Ae. aegypti*, for example (shown alongside all other

**Fig. 2** Sex-specific and species-specific auditory transduction in mosquito flagellar ears. **a** Median receiver slope stiffness and nerve responses in response to flagellar displacements for female (red) and male (blue) *Ae. aegypti* (AEG), *Cx. quinquefasciatus* (QUI) and *An. gambiae* (GAM). Solid lines show the best fit of the single transducer population model to the stiffness data. Error bars represent ± SEM. Sample sizes: *Ae. aegypti* females = 21; *Ae. aegypti* males = 18; *Cx. quinquefasciatus* females = 17; *Cx. quinquefasciatus* males = 15; *An. gambiae* females = 18; *An. gambiae* males = 11. Supplementary Figure 4 shows equivalent responses in terms of rotational stiffness in response to angular deflections. **b** Median antennal nerve response magnitudes in response to flagellar displacements; dotted lines represent the best ion channel open probability fit for median values of the antennal nerve response for female and male *Cx. quinquefasciatus* (QUI) mosquitoes. Error bars represent ± SEM. Sample sizes: *Cx. quinquefasciatus* females = 17; *Cx. quinquefasciatus* males = 15. **c** Semi-log plots of the proportion of maximum antennal nerve response produced for increasing flagellar displacements for female and male *Ae. aegypti* (AEG), *Cx. quinquefasciatus* (QUI) and *An. gambiae* (GAM), respectively; small, hollow points represent data for individual mosquitoes whilst larger, solid points represent median values for a group. For each species the displacement required to produce half-maximal CAP responses in females is almost 10 times as large as for males, for whom 50% saturation occurs within ~250 nm (see also Table 2). Sample sizes: *Ae. aegypti* females = 21; *Ae. aegypti* males = 18; *Cx. quinquefasciatus* females = 17; *Cx. quinquefasciatus* males = 15; *An. gambiae* females = 18; *An. gambiae* males = 11

### Table 2 Parameter values for fits of the single transducer population gating spring model

|  | AEG ♀ | QUI ♀ | GAM ♀ | AEG ♂ | QUI ♂ | GAM ♂ |
|---|---|---|---|---|---|---|
| Channel number, $N$ | 544.6*** | 801.8*** | 764.1*** | 106.9*** | 73.8*** | 60.5*** |
| Channel gating force, $z$ (fN) | 23.7***,† | 22.1***,† | 34.4***,† | 51.0*** | 56.4*** | 144.9*** |
| $K_{INFINITY}$ (μN/m) | 93.8***,†† | 103.1***,†† | 147.8**,†† | 178.3***,†† | 190.6***,†† | 283.8**,†† |
| $K_{STEADY}$ (μN/m) | 82.4***,† | 85.7***,††† | 112.7***,††† | 147.7***,† | 159.1***,† | 208.1***,† |
| $K_{GS}$ (μN/m) | 11.5***,††† | 17.3**,††† | 35.1***,††† | 30.5***,††† | 31.5**,††† | 75.8***,††† |
| Extent of nonlinearity | 0.202*,†† | 0.235*,†† | 0.379***,†† | 0.096***,††† | 0.076*,†† | 0.276***,††† |
| $CAP_{50}$ (nm) | 1079*** | 741*,† | 1425***,† | 180*** | 145* | 135*** |
| $\lambda_{90}$ (nm/mdeg) | 1023.9/58.7 | 1098.3/62.9 | 705.0/40.4 | 476.1/27.3 | 430.9/24.7 | 167.6/9.6 |

Function fit values for a single transducer population fit to median stiffness values (unified to a measurement height of 1 mm above flagellar base, see Supplementary Table 1 for flagellar lengths for each sex and species) for female and male *Ae. aegypti* (AEG), *Cx. quinquefasciatus* (QUI) and *An. gambiae* (GAM). $N$ is the estimated number of transducer channels and $z$ is the force change following the opening of a transducer channel for one gating spring. Here, $K_{INFINITY}$ is the asymptotic flagellar stiffness at large displacements, whilst $K_{STEADY}$ is the combined linear elasticity of the flagellar joint and neurons of the flagellar receiver. $K_{GS}$ (the gating spring stiffness) was calculated as $K_{INFINITY} - K_{STEADY}$ and so provides information on transducer module mechanical integrity. The extent of nonlinearity indicates the degree to which the system is nonlinear. $CAP_{50}$ values refer to the minimum flagellar displacement required to produce an antennal nerve response of 50% of the maximum response. $\lambda_{90}$ values refer to the displacement at which 90% of the transducers are predicted to be open. Significant differences between conspecific female and male mosquitoes are starred (ANOVA on ranks; *$p < 0.05$; **$p < 0.01$; ***$p < 0.001$). Significant differences between mosquitoes of the same sex but of different species are also highlighted (ANOVA on ranks; †$p < 0.05$; ††$p < 0.01$; †††$p < 0.001$)

mosquito groups in Supplementary Figure 3), settled around the level of the asymptotic stiffness, $K_{INFINITY}$, which is consistent with a near-complete block of transducers[34]. Residual nonlinearities close to the receivers' resting positions partly remain after pymetrozine exposure; these may reflect increased mechanical noise from the disabled transducers or additional, transduction-independent system nonlinearities. Application of pymetrozine also led to complete loss of mechanically evoked CAP responses.

**Sex-specific and species-specific efferent innervation of JO.** The above biophysical analyses uncovered substantial sexual dimorphisms of JO auditory function in all three mosquito species tested. To see if, and how, these functional differences associated with structural differences, we studied JO functional neuroanatomy by way of immunohistochemistry. Specific focus was placed on the JOs' efferent innervation patterns due to their potential roles in modulating auditory sensitivity.

We began by confirming that the efferent network recently described for the ears of *Cx. quinquefasciatus*[23] was also present in the ears of the two other mosquito species studied here. Efferent terminals, stained using the presynaptic marker 3C11 (anti-synapsin), were present in all three mosquito species (Fig. 3).

Different levels of sexual dimorphism were observed in the specific distribution, and cellular location, of the efferent terminal network across the mosquito species. Males from all species showed an extensive network of efferent fibres innervating different JO regions, including the area directly underneath the basal plate, the base of auditory cilia, the neuronal somata and the auditory nerve (Fig. 3a–d).

In contrast, female innervation was restricted to the base of the auditory cilia and the somata region in *Ae. aegypti* and *Cx. quinquefasciatus* females (Fig. 3e, g, h). Most notably, JO size and complexity of efferent innervation appear dramatically reduced in female *An. gambiae* (Fig. 3f, i).

**Disruption of nerve signalling leads to male-specific SOs.** Substantial differences in auditory efferent innervation thus exist between male and female mosquitoes from each species. Given these differences and the male-specific nature of SOs, we tested whether the higher degree of neuronal complexity observed in the JO of male mosquitoes is linked to the generation, or control, of SOs.

We first studied flagellar displacements and the responsiveness of the antennal nerve to mechanical stimulation in male mosquitoes before and after the onset of SOs. Flagellar displacement amplitudes increased by several orders of magnitude whilst exhibiting SOs and mechanically evoked nerve responses persisted. Figure 4a (left) shows a male mosquito which displayed spontaneous SOs during an experimental series with CAP recordings from the antennal nerve showing a strong, phase-locked response at twice the frequency of the flagellar oscillation[35]; thus, mechanosensory transduction remained intact during SOs.

We then addressed the functional contributions of efferent innervation by quantifying the specific auditory phenotypes that result from systemic blocks of afferent/efferent signalling. We utilised two pharmacologically and conceptually different

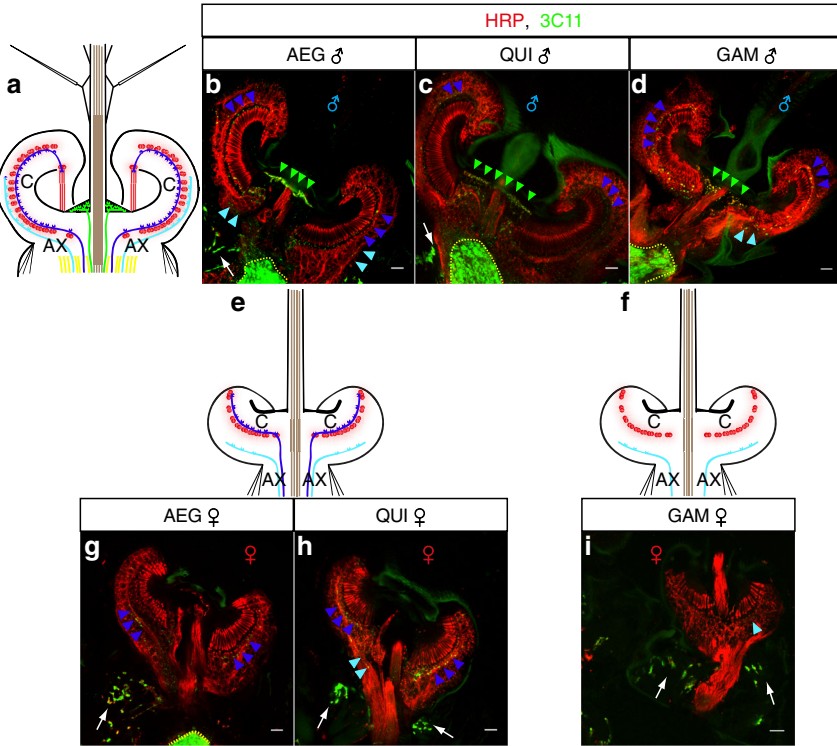

**Fig. 3** Sexual dimorphisms in the auditory efferent innervation of all three mosquito species JO horizontal sections were stained with the presynaptic marker 3C11 (anti-synapsin, green) to label auditory efferent fibres[23] and counterstained with the neuronal marker anti-HRP (red) for *Ae. aegypti* (AEG), *Cx. quinquefasciatus* (QUI) and *An. gambiae* (GAM). **a**, **e**, **f** Sketches of the three different patterns of efferent innervation observed. Efferent fibres are classified according to the region innervated: underneath the basal plate (green); base of auditory cilia (dark blue); somata (light blue); auditory nerve (yellow). The coding colour also applies to the arrowheads in **b**–**i**. AX axons, C auditory cilia. Modified from refs [7,8,23]. **a**–**d** Male mosquito JO of all three species present an extensive efferent innervation pattern—as revealed by 3C11 staining—in the basal plate (green arrowheads), base of auditory cilia (dark blue arrowheads), intermingled among somata (light blue arrowheads) and in the auditory nerve (yellow dash line). **e**, **g**, **h** In AEG and QUI females, the efferent fibres innervate the base of the auditory cilia (dark blue arrowheads) and somata region (light blue arrowheads). **f**, **i** Efferent innervation in GAM females is limited to dispersed punctae intermingled among the somata (light blue arrowhead). 3C11 also stains motoneuronal innervation of muscles in the scape (arrow). Scale bar: 10 µm. Supplementary Figure 5 contains single channel, as well as merged, images

strategies: injection of either tetrodotoxin (TTX) or tetanus toxin (TeNT). TTX blocks voltage-gated sodium channels[36], leading to a loss of all action potential-based signalling. TeNT however binds to presynaptic membranes and blocks neurotransmitter release[37], resulting in a loss of signalling across chemical synapses. Both interventions should therefore disrupt all afferent/efferent signalling pathways between the mosquito JO and brain which involve action potential-dependent or synapse-dependent signalling.

Male flagellar receivers from all species showed the same behaviour in response to both TTX and TeNT injections: large-amplitude SOs (Fig. 4a, right; Fig. 4b, right), which closely resembled spontaneous SOs. In each case, the frequencies of the pharmacologically induced SOs were lower than the flagellar best frequencies of the ringer-injected control state (Fig. 4b, right). Subsequent injection of the transduction-blocker pymetrozine abolished SOs in all cases (Fig. 4a, right).

Quantification of flagellar power gains during the SOs revealed the extent of auditory amplification across the three species. Power gains rose by >10-fold in males of *Ae. aegypti*, by >100-fold in males of *Cx. quinquefasciatus* and by ~10,000-fold in males of *An. gambiae*, where they reached values up to ~45,000$k_{B}T$ following TeNT injection (Figure 4c and Table 3). In contrast to males, the flagellar receivers of *Ae. aegypti* and *An. gambiae* females did not show any statistically significant response to TTX or TeNT injection (Fig. 4b, left). In *Cx. quinquefasciatus* females,

power gain levels rose post-injection by ~2-fold to ~23$k_{B}T$ (Fig. 4c and Table 3); this increase in power gain is orders of magnitudes smaller than for conspecific males however, as can be seen from the corresponding free fluctuation data (Supplementary Figure 2c). Comparative TTX injections into *Drosophila* produced no change in the antennal free fluctuations (Supplementary Figure 2d), in agreement with previous reports of a lack of efferent innervation in the *Drosophila* JO[38]. Injection of pymetrozine, as before, led to the flagellar receivers of all mosquitoes tested (including those displaying SOs) becoming similar to their passive states.

We then explored the responses of male ears that displayed spontaneous SOs to external stimulation. We recorded mechanical and electrical responses to pure tones close to the SO frequency (~361 Hz in Fig. 5a). Without external stimulation, ears displayed continuous phase-locked nerve responses at twice the SO frequency (Fig. 5a), consistent with the frequency doubling reported for Dipteran antennal ears[25]; the power spectral density (PSD) of corresponding receiver vibrations shows a single major peak at the SO frequency. When playing a tone at ~16 Hz below the receiver's SO frequency, flagellar vibrations show signs of waveform interference, which are also reflected in the nerve responses; the corresponding PSD now shows two separate peaks, one at the SO frequency and one at the (lower) stimulus frequency, that is, the stimulus tone has failed to entrain the spontaneous SOs.

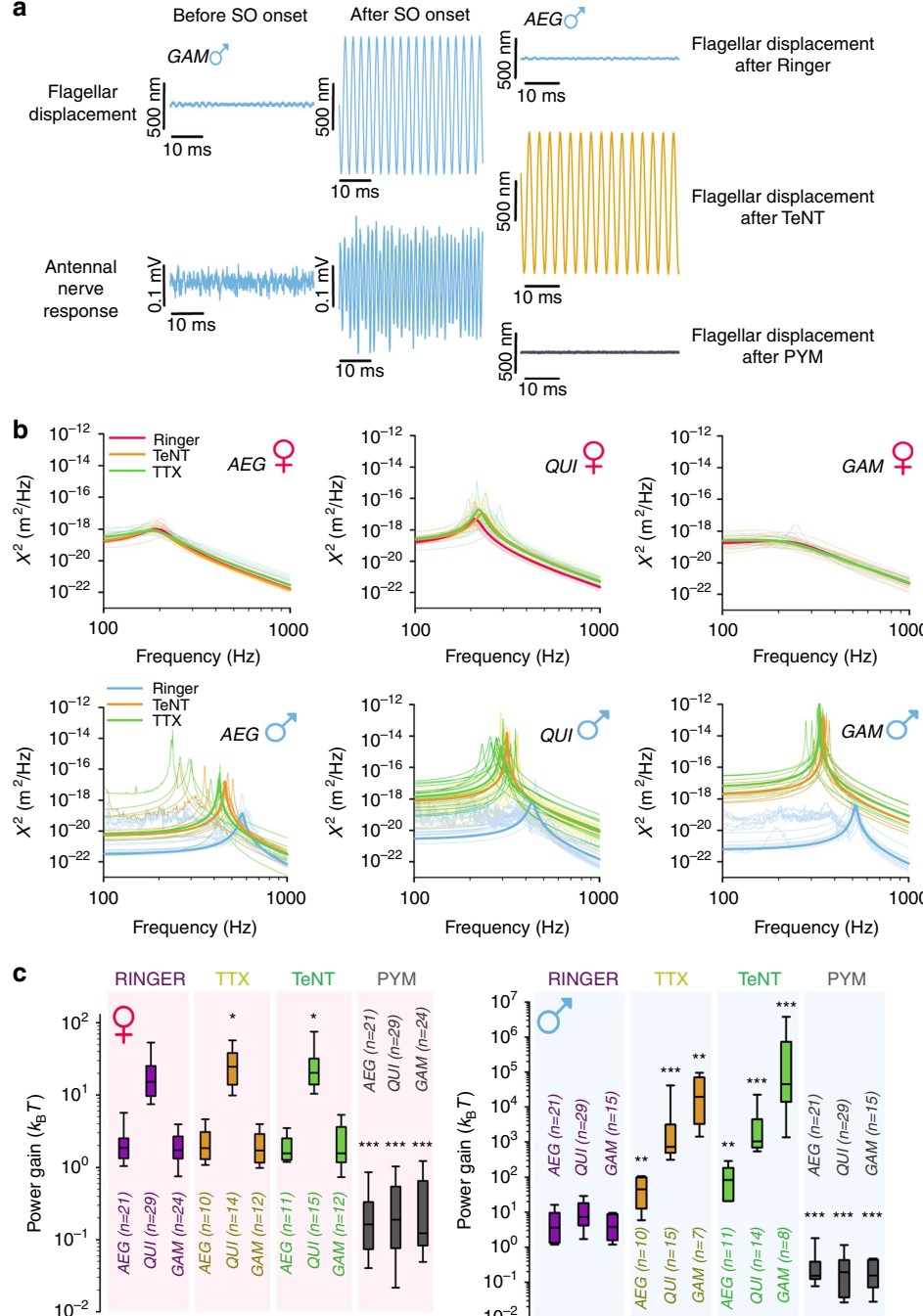

**Fig. 4** Disrupting the afferent/efferent control loop produces large self-sustained oscillations (SOs) in males. **a** (Left) Unstimulated flagellar displacements and corresponding antennal nerve responses before and after the onset of spontaneous SOs for an *An. gambiae* (GAM) male. (Right) Unstimulated flagellar displacements following ringer, TeNT and pymetrozine injection for an *Ae. aegypti* (AEG) male. See Supplementary Table 2 for comparisons between spontaneous and induced SOs. **b** Power spectral densities (PSDs) from harmonic oscillator fits to free receiver fluctuations of female and male *Ae. aegypti* (AEG), *Cx. quinquefasciatus* (QUI) and *An. gambiae* (GAM) flagella in three separate states: after Ringer injection, after TTX injection and after TeNT injection. Prominent solid lines represent fits created from median parameter values (i.e. median values from a population), whilst shaded lines represent damped harmonic oscillator fits for individual mosquitoes. **c** Power gain estimates for female (left) and male (right) *Ae. aegypti* (AEG), *Cx. quinquefasciatus* (QUI) and *An. gambiae* (GAM) after Ringer injection, TTX injection, TeNT injection or pymetrozine (PYM) injection. Significant differences between injection states within a population are starred (repeated-measures ANOVA on ranks; *$p < 0.05$; **$p < 0.01$; ***$p < 0.001$). Centre line, median; box limits, lower and upper quartiles; whiskers, 5th and 95th percentiles. Sample sizes (after Ringer/after TTX/after TeNT/after pymetrozine): *Ae. aegypti* females = 21/10/11/21; *Ae. aegypti* males = 21/10/11/21; *Cx. quinquefasciatus* females = 29/14/15/29; *Cx. quinquefasciatus* males = 29/15/14/29; *An. gambiae* females = 24/12/12/24; *An. gambiae* males = 15/7/8/15

**Table 3 Principal parameters from free fluctuation analysis after compound injection**

|  | AEG ♀ | AEG ♂ | QUI ♀ | QUI ♂ | GAM ♀ | GAM ♂ |
|---|---|---|---|---|---|---|
| TTX+ state |  |  |  |  |  |  |
| Sample size | 10 | 10 | 14 | 15 | 12 | 7 |
| Best frequency (Hz) | 196.90 (3.09) | 445.33* (29.55) | 210.58 (4.54) | 317.08*** (7.67) | 215.00 (6.64) | 339.63** (6.51) |
| Tuning sharpness $Q$ | 2.76 (0.45) | 68.13** (58.90) | 16.01* (4.89) | 156.88** (290.01) | 1.21 (0.06) | 245.76** (945.39) |
| TeNT+ state |  |  |  |  |  |  |
| Sample size | 11 | 11 | 15 | 14 | 12 | 8 |
| Best frequency (Hz) | 199.27 (2.05) | 353.04** (25.09) | 222.44 (6.39) | 301.21*** (7.32) | 218.70 (3.51) | 327.01*** (8.90) |
| Tuning sharpness $Q$ | 2.55 (0.24) | 96.14*** (557.70) | 14.07* (6.98) | 309.65*** (162.60) | 0.99 (0.16) | 3034.38*** (1374.67) |
| Combined male induced SO state |  |  |  |  |  |  |
| Sample size | – | 21 | – | 29 | – | 15 |
| Best frequency (Hz) | – | 384.43 (20.45) | – | 311.23 (5.52) | – | 332.04 (5.81) |
| Tuning sharpness $Q$ | – | 93.05 (296.39) | – | 239.28 (166.47) | – | 355.79 (895.09) |
| Power gain Ringer ($k_{\mathrm{B}}T$) | 1.84 (0.52) | 3.61 (1.68) | 15.35 (4.88) | 7.47 (5.48) | 1.72 (0.24) | 3.78 (0.83) |
| Power gain TTX ($k_{\mathrm{B}}T$) | 1.56 (0.40) | 45.43** (14.62) | 24.63* (4.38) | 722.72*** (6004.67) | 1.42 (0.31) | 19280.68** (14592.23) |
| Power gain TeNT ($k_{\mathrm{B}}T$) | 1.56 (0.25) | 66.57** (99.16) | 21.89* (9.20) | 1078.9*** (2055.62) | 1.55 (0.45) | 44934.32*** (469402.29) |
| Power gain-induced SO state ($k_{\mathrm{B}}T$) | – | 49.70 (19.87) | – | 1041.56 (3218.82) | – | 40134.86 (254561.08) |
| Power gain pymetrozine ($k_{\mathrm{B}}T$) | 0.15*** (0.08) | 0.15*** (0.16) | 0.17*** (0.15) | 0.19*** (0.07) | 0.12*** (0.12) | 0.15*** (0.07) |

Median values obtained from harmonic oscillator fits for female and male *Ae. aegypti* (AEG), *Cx. quinquefasciatus* (QUI) and *An. gambiae* (GAM) following TTX or TeNT injection (standard errors are given in brackets); these include the best frequency and tuning sharpness ($Q$) of the flagellum. Combined median values including both TTX and TeNT injected states are shown to present data for all males exhibiting induced self-sustained oscillations (SOs). Median values of the power gain in four injection states (after Ringer, TTX, TeNT or pymetrozine injection), as well as a value for the combined induced state (including both TTX and TeNT), are given with corresponding standard errors. Significant differences in best frequency/$Q$/power gain between the control (Ringer) state and any other state for a specific mosquito group are starred (repeated-measures ANOVA on ranks; *$p < 0.05$; **$p < 0.01$; ***$p < 0.001$). No significant differences in best frequency/$Q$/power gain between the TTX exposed and TeNT exposed states for a specific mosquito group were calculated (ANOVA on ranks with a significance level of $p < 0.05$)

Increasing the stimulus tone frequency further to a value of only ~6 Hz below the SO frequency, however, leads to a sudden change of behaviour. Obvious signs of waveform interference disappear from both the flagellar vibrations and nerve response, with the corresponding PSD shows only a single (considerably increased) major peak at the frequency of the external stimulus. This indicates that the receiver SOs have been entrained by the external stimulus. This entrainment response is repeated for stimulus frequencies of ~4 and ~14 Hz above the SO frequency. Increasing the frequency of the external stimulus further to a value of ~24 Hz above the SO frequency, however, again results in entrainment failure; flagellar vibrations and nerve responses again show signs of waveform interference and the corresponding PSD contains two major peaks, one at the SO frequency and one at the (now higher) stimulus frequency. Figure 5b demonstrates the narrowness of the frequency range where entrainment was possible for individual mosquitoes, with a maximum range of 30 Hz (i.e. ±15 Hz as compared to the SO frequency) being identified.

## Discussion

Prior studies[9,17] have provided evidence suggesting that mosquito flagellar ears are active mechanosensors, expected to operate away from thermal equilibrium. Thus, like the hair bundles of vertebrate inner ear hair cells[39] and the *Drosophila* antennal ear[28], they are thought to inject energy into mechanically evoked motions of their stimulus receivers; direct demonstration of such power gain, however, has not yet been reported. We here provide a systematic framework for the analytical and quantitative dissection of mosquito hearing. This has uncovered several novel insights into the function, sexual dimorphism and evolution of mosquito ears and has suggested new interpretations of previously reported phenomena.

One major finding is that the ears of males and females of all three species displayed power gain. Baseline (median) power gain values for the quiescent receivers (i.e. receivers not undergoing SOs) from all three mosquito species ranged between ~2–6$k_{\mathrm{B}}T$. This is surprisingly similar to values reported for *Drosophila* controls of ~4–8$k_{\mathrm{B}}T$[28,40] given the ~20 times (females) or ~40 times (males) larger size of the mosquito JO.

As mentioned previously, however, the neurons of the mosquito JO are grouped in prongs. Prongs are radially arranged cuticular processes, to which numbers of neurons are attached. This arrangement is thought to be the structural basis for the mosquitoes' exquisite ability to localise a sound source. Male JOs possess ~70 prongs, which would, based on purely structural considerations, correspond to a ~5° angular resolution[14]. One particular question that has remained unclear is the degree of mechanical separation between neighbouring prongs. In other words: If the flagellum is displaced within one plane, does the excitation spread across multiple prongs or does it remain restricted to the prongs within the plane of flagellar displacement? Here, our data can at least provide first circumstantial evidence indicating that the prongs appear to be mechanically largely separated from each other.

If there was a vectorial spread of neuronal excitation across the various prongs of the male JO, then the proportion of effectively responding neurons would be >50% for each plane of stimulation. This would not only blur spatial resolution and impair sound source localisation but also imply that the energy contributions per neuron would be at least ~20 times lower than those of *Drosophila*. If however one assumes mechanical separation, then our data would represent contributions from neurons between two anatomically opposing prongs only.

There are ~16,000 neurons divided into ~70 prongs in the male *Ae. aegypti* JO[23]. The power gain values of this study would thus reflect the contributions of ~460 neurons (in two opposing prongs). In *Drosophila* (due to differences in functional anatomy),

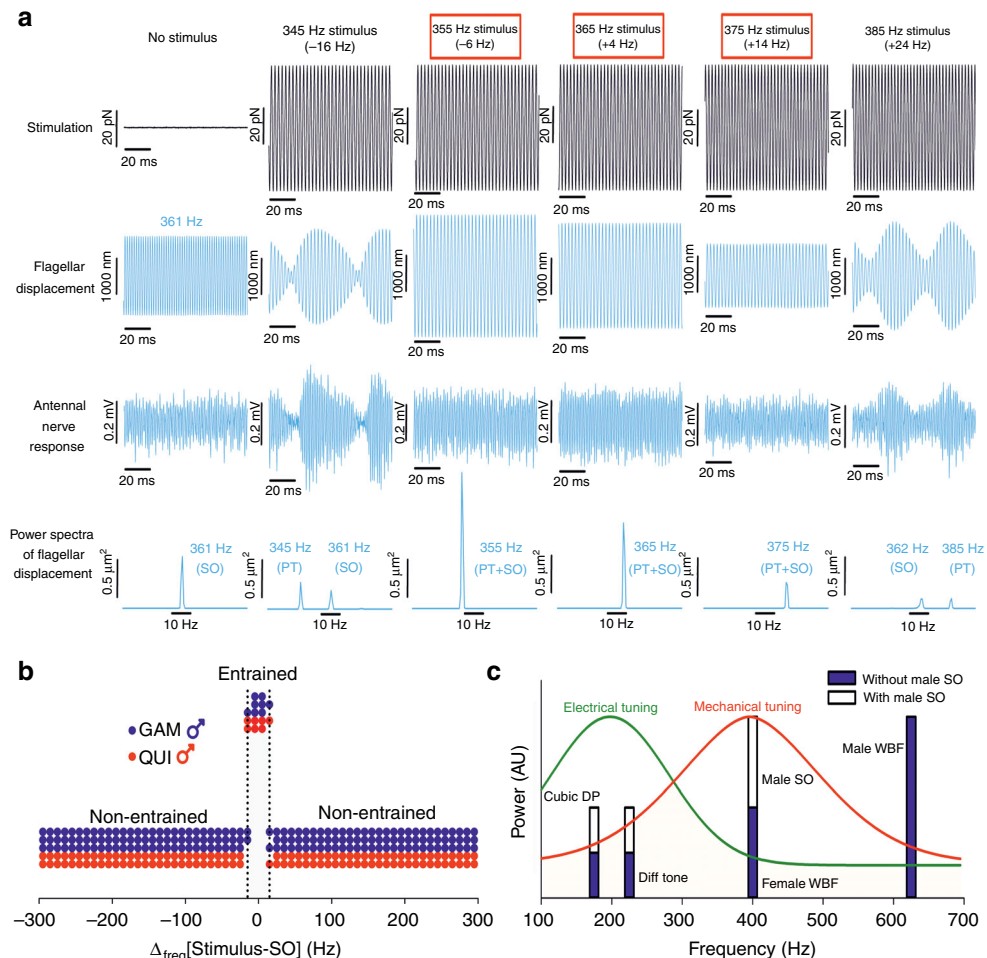

**Fig. 5** Extents of amplification and frequency-specific response behaviour in spontaneously oscillating male receivers. **a** Flagellar displacements (2nd row from the top) and antennal nerve responses (3rd row) to pure tone stimulation (1st row) for an *An. gambiae* male following the onset of spontaneous SOs. The SO frequency was measured as 361 Hz. Red boxes highlight stimulus frequencies for which entrainment was judged to occur. Power spectra of flagellar displacements (4th row) for each stimulus frequency are included to visualise frequency-dependent changes in the receiver's entrainment behaviour. **b** Frequency range over which individual male *An. gambiae* (GAM, blue) or *Cx. quinquefasciatus* (QUI, red) exhibiting spontaneous SOs entrained to pure tone stimulation of different frequencies. The frequency range is represented as the difference between the frequency of the pure tone stimulus itself and the best frequency of the SO. Individual data points indicate the entrainment status of each mosquito at each frequency measured, which ranged from ±300 Hz the difference between the pure tone stimulus and the SO best frequency with 10 Hz increments between each tone used. The shaded area, which covers a range of 30 Hz (i.e. ±15 Hz), represents the maximum region over which SO entrainment to the stimulus took place. *Cx. quinquefasciatus* males = 2; *An. gambiae* males = 3. There is no data for *Ae. aegypti* males as they did not show spontaneous SOs under our experimental conditions. **c** Diagrammatic representation of hypothesised effect of the male SO on electrophysiological responses of male JONs in the presence of the female flight tone. Power units and scale of responses are arbitrary. Solid green line represents electrical tuning for male mosquitoes, whilst the solid red line represents male mechanical tuning. SO=self-sustained oscillation, WBF=wingbeat frequency, cubic distortion=product difference between twice female WBF and male WBF, difference tone=difference between male and female WBFs

all JO neurons (~480 in total[41]) are likely to contribute; thus, the total number of contributing neurons would be roughly the same, explaining the almost identical levels of power gain. This may also indicate that the levels of baseline energy injection are a conserved feature across the scolopidia of Dipteran insects.

The extent of energy injection between male and female mosquitoes was broadly similar across all three species tested, although neuronal numbers are reported to differ by a factor of ~2. Again, the fact that the neurons in the female JO are arranged into fewer prongs is likely to contribute to the equal levels of male and female power gain. These relations may reflect an evolutionary trade-off sacrificing angular resolution for absolute sensitivity. Indeed, female mosquito ears demonstrated comparative sensitivity to quiescent male ears in multiple tests of auditory

function, suggesting that the auditory world of female mosquitoes is richer than currently appreciated.

From a sensory ecological perspective, it seems noteworthy here that bidirectional acoustic interactions have been reported between males and females flying in couple[10], perhaps hinting at a female choice component in mosquito mating[42]. Also, roles of audition beyond mating might include predator avoidance (in both males and females)[43] or host finding (in females). Corresponding phonotactic responses related to frog calls have indeed been reported for females of frog-biting mosquito species[44,45], including *Culex* spp[46]. This possibly explains why *Cx. quinquefasciatus* was the only species in our study where female baseline auditory amplification exceeded that of males.

Male flagellar receivers exhibiting SOs are distinct however; their energy content rose to values ~4 orders of magnitude above mosquito baseline levels, ~3 orders of magnitude above pharmacologically induced *Drosophila* SOs[28] and ~2 orders of magnitude above estimated limits for the transducer-based active process in vertebrate hair cells[47]. This may imply differences in underlying amplificatory mechanisms, potentially involving the two identified mosquito orthologues of the mammalian outer hair cell motor protein Prestin[48], though myosins and dyneins could also be possible candidates. Although the *Drosophila* Prestin orthologue does not seem to contribute to mechanical feedback amplification[49], this question still awaits experimental clarification in mosquitoes.

Our analyses of auditory transducers uncovered substantial sex-specific and species-specific differences (Table 2), suggesting that the molecular evolution of auditory transducer modules lies at the heart of variations in mosquito auditory function. We also discovered fundamental commonalities between auditory transduction in mosquitoes, fruit flies[25,28] and vertebrate hair cells[24,50]; these include directly gated transducer modules and transducer-based mechanical feedback amplifiers, which provide power gain for mosquito hearing.

We focused our first quantitative analysis of auditory transducer gating in mosquitoes on small deflections around the flagellar resting position. This approach ensured we (i) analysed and compared only the most sensitive population of transducers for each sex and species, respectively, and (ii) could use a simpler formulation of the gating spring model previously utilised to analyse small deflections of the *Drosophila* ear[25]. This model assumes only a single, homogenous transducer population. Research in the *Drosophila* JO has identified additional, functionally distinct, mechanotransducer populations which contribute to mechanosensory behaviours beyond audition[51,52] and differ in their molecular make-up[33,53]. The most sensitive (auditory) population of transducers, however, appears to contribute over-proportionately to tuning and amplification[54,55].

Future research could focus on identifying further mechanotransducer populations in mosquitoes as the data presented here also suggests the existence of functionally distinct populations, in agreement with recent reports for *Cx. pipiens* males[43]. Intriguingly, our data show that one of the main differences between male and female ears is the gating properties of their auditory transducers: the males of all species had transducer modules with (i) a greater total gating spring stiffness, $K_{GS}$, (ii) larger single channel gating forces, $z$, and (iii) smaller numbers of predicted transducer channels, $N$, than conspecific females (Table 2).

These sex-specific variations match theoretical expectations for transducer populations of different sensitivities[56] and are also in close agreement with differences found experimentally between sensitive (auditory) and insensitive (wind/gravity) transducers in the *Drosophila* ear, where they have also been linked to a differential molecular make-up[33]. In addition to possible molecular specialisations, variations in transducer geometry (which modify force transmission between the antennal receiver and different JO cilia) could further contribute to the differences observed in both *Drosophila* and mosquitoes.

Irrespective of the particular mechanisms however, in mosquitoes the ears of all males possess more sensitive transducers than conspecific females, suggesting particular ecological specialisations. It seems plausible that the male-specific behaviour of detecting, locating and chasing a female flying by is the ecological context of this transducer variation. Further research is needed to unravel the full extent and functional relevance of sex-specific auditory adaptations in mosquitoes. It is unclear whether specialisation is restricted to particular classes of auditory neurons, such as the most sensitive ones or spiking/non-spiking ones[43]; the greatly diminished CAP amplitudes found in Anopheline females could hint at a specific reduction of spiking neurons. The functional investigation of these extensive sexual dimorphisms, however, has just started.

On the species level, both sexes of the two culicine species (*Ae. aegypti* and *Cx. quinquefasciatus*) had a lower total gating spring stiffness, $K_{GS}$, and smaller single channel gating forces, $z$, than their corresponding sex of the anopheline species, *An. gambiae* (Table 2). Thus, both intersexual and interspecific differences were found in the mosquitoes' auditory transducer populations. For example, transducer working ranges were significantly smaller in males than in females. Auditory transducers of male *An. gambiae* were predicted to be 90% open ($\lambda_{90}$; ref. [57]; Table 2) when their flagellar receiver was deflected only 168 nm away from its resting position; the receivers of conspecific females needed to be moved by ~4 times as much (705 nm) in order to reach the same open probability.

Conversion of the $\lambda_{90}$ displacements into angular deflections (Table 2) facilitates comparisons within this study as well as with previously published sensitivity estimates for mosquitoes[9] or vertebrate hair cells[58]. In angular terms, the $\lambda_{90}$ sensitivity of *An. gambiae* males represents a deflection of <0.01° and those of the females of ~0.04°. For comparison, equivalent deflections for the mechanosensory hair bundles of vertebrate inner ear hair cells are >100 times larger, ranging from 1° to 6°[58].

Our findings on the effects of blocking JO efferent innervation raise the question of the neurobiological and behavioural roles of SOs, which so far remain unclear. Given that (i) pharmacologically induced and spontaneously occurring SOs are only found in males, (ii) the auditory nerve responds to the SOs (Fig. 4a), (iii) the nerves of ears undergoing SOs remain sensitive to additional stimulation (Fig. 5a) and (iv) pharmacologically induced and spontaneously occurring SOs are highly similar to each other, SOs are likely to represent a key feature, rather than a pathological state, of the male hearing mechanism. We suggest that SOs are controlled, and suppressed, by the efferent innervation of the male ear; thus, blocking efferent signalling releases this suppression. Further research is required to explore the specific roles of various neurotransmitters and synaptic transmission sites identified within the mosquito JO[23].

Here, SOs behaved like powerful, narrowband lock-in amplifiers, entrained only by pure tones around their oscillation frequency (Fig. 5a, b). SO frequencies were similar to previously reported female wingbeat frequencies[11–13]. SOs could therefore act as highly specific amplifiers of faint female flight tones. This scenario is relevant in the context of the distortion product (DP)-based communication system previously proposed for mosquitoes[59], particularly for conspecific, intersexual communication within swarms. SOs might be part of an enhanced sensing landscape, as has been proposed as an emergent property of mobile animal groups such as mosquito swarms[60]. Indeed, it has been suggested that a male mosquito's own wingbeat is a vital constituent of signal detection in his auditory system[19].

DP-based communication relies on nonlinear mixing between two pure tones (e.g. male and female wing beats), which leads to the generation of additional, mathematically predictable, tones[61]. For a flying male mosquito, one of these tones (his own wingbeat) is inevitable and loud; in tethered flying *Drosophila*, it has been found to be large enough to saturate all JO neurons[62]. The second tone (the female wingbeat), however, is faint in comparison.

We hypothesise that the male's strategy is to create an internal simulation of a flying female of sufficient amplitude to generate a small DP. Every additional (external) energy injection into this specific frequency band, such as that provided by a nearby female, will then modulate and increase the DP. Here three things are

particularly relevant: (i) SOs can match (entrain) their frequency to an external stimulus (e.g. a female wingbeat) within a range of ~±15 Hz around the SO's unforced natural frequency (Fig. 5a, b), (ii) mismatches between SO and external stimulus frequency lead to significant waveform interferences in both flagellar oscillations and corresponding nerve responses (Fig. 5a) and (iii) efferent modulation[23] might be able to fine-tune the SO's natural frequency, thus extending the operational range of the SO-based lock-in amplifier.

Taken together, such an auditory system would enable the male to detect, and amplify, a faint female flight tone by locking into the female wingbeat frequency and using low-frequency DPs of the amplified female flight tone and his own wingbeat frequency. As reported before[12,63], the nerves of all males tested here were most sensitive to stimulus frequencies around these predicted low-frequency DPs. By using DPs rather than the original flight tones, males could turn the apparent noise of their own wingbeat into a signal amplifier (Fig. 5c). The ears of male mosquitoes would thus form a biological equivalent of a superheterodyne receiver, or superhet; virtually all modern radios operate according to the superhet principle[64]. Future studies will have to further test this proposal, especially for naturally occurring levels of male and female wing beats.

Our findings recommend strategies that target hearing and acoustic communication, which are essential components of courtship behaviour in all major mosquito disease vectors, as promising novel routes for vector control[3,65]. Targeting this shared sensory ecological bottleneck (whether through novel insecticides, acoustic traps or other innovative methods) could help to overcome limitations of current insecticidal approaches. For example, insecticide-treated bed nets primarily target mosquitoes with distinct dusk and dawn activity patterns (An. gambiae)[15], but fail to capture more ecologically flexible species with less strict patterns of circadian behaviour (Ae. aegypti)[66]. Considering the substantial investments of energy made by male ears, a potential circadian control over auditory energy expenditure (modulated by efferent innervation of the male JO) is here an intriguing possibility. The different diurnal activity rhythms of the three species studied would offer an ideal opportunity to study this question.

## Methods

**Mosquito rearing**. All Ae. aegypti, Cx. quinquefasciatus (Muheza) and An. gambiae (Kisumu) used for experiments were provided by Shahida Begum from the London School of Hygiene and Tropical Medicine. All mosquitoes were reared using a 12 h:12 h light–dark cycle at 26 °C and 75% relative humidity and were fed a 10% glucose mixture. Horse blood feeding, where appropriate, was completed by a trained research assistant using the Hemotek system (Discovery Workshops, Accrington). All mosquitoes used for experiments (unless otherwise noted) were between 3 and 8 days old. No randomisation of mosquitoes or blinding of investigators was done for experiments. Whilst male Ae. aegypti and Cx. quinquefasciatus antennal fibrillae are permanently erect, those of male An. gambiae are erect only during strict circadian time windows associated with swarming behaviour[67]. All recordings were made within a 2 h time window beginning 1 h after light onset—thus, male An. gambiae fibrillae were not erect throughout these experiments.

**Laser Doppler vibrometry preparation**. Mosquitoes were first glued to a Teflon rod using blue-light-cured dental glue (as has been reported for Drosophila melanogaster[33]). The glue was then spread across other body parts to minimise disturbances caused by movements of the mosquito (with attention given to not obstructing flagellar motion and not obscuring abdominal or thoracic spiracles). The left flagellum was then adhered to the head and glue was applied between the pedicels; leaving only the right flagellum free to move.

The rod holding the mosquito was placed in a micromanipulator atop a vibration isolation table, with the mosquito facing the laser Doppler vibrometer at a 90° angle. Different laser focus points were chosen for male and female mosquitoes based upon preliminary testing in order to minimise disturbances; for males, the second flagellomere from the flagellum tip was used, whilst for females the third flagellomere from the tip was utilised. All recordings used a PSV-400 laser Doppler vibrometer (Polytec) with an OFV-70 close up unit and a DD-500 displacement

decoder. Figure 1a shows a sketch of the laser Doppler vibrometry (LDV) experimental paradigm. All measurements were taken in a temperature-controlled room (22 °C) within a time window of 0 to 3 h following light onset.

**CO$_2$ sedation experiments**. Mosquitoes were mounted as described above before being placed inside a rectangular steel chamber ($6 \times 6 \times 2.5$ cm$^3$), as has been reported for D. melanogaster[40]. This chamber was positioned opposite the laser Doppler Vibrometer and held in a micromanipulator. One side of the chamber contained a glass window which allowed for recording flagellar vibrations from the mounted mosquito.

A free fluctuation recording was taken prior to CO$_2$ exposure, with a plastic case ($3.5 \times 2.5 \times 3$ cm$^3$) being put on top of the mosquito which prevented rapid decreases in CO$_2$ concentration following gas flow cessation. CO$_2$ was then allowed to flood the chamber through a porous membrane floor for 1 min at a constant flow rate of 3 l/min (maintained using a flow regulator (Flowbuddy, Flystuff)). Free fluctuation recordings were taken in a loop throughout this time to investigate the mosquito's active hearing system. After this, the CO$_2$ flow was halted and a free fluctuation recording of the passive mosquito flagellum was taken. The mosquito was then given 5 min to recover before a final free fluctuation was recorded.

Mosquitoes which did not show signs of recovery from the CO$_2$ sedation were excluded from the final analysis. This recovery was judged by determining the best frequency and velocity amplitude of the mosquito flagellum (with relevant analytical procedures explained below) as compared to that in the pre-CO$_2$-exposed state, with changes of >20% from this original state being considered grounds for exclusion. These recovery criteria were adopted for all experiments utilising CO$_2$ sedation or electrostatic stimulation.

**Compound injection procedure**. Five micromolar TTX, 20 nM TeNT and 100 μM pymetrozine solutions (all of which were diluted from stock solution using Ringer[68]) were prepared for use in injection experiments. Sharpened micro-capillaries were filled with the appropriate solutions (including a ringer control). The tip of these micro-capillaries was inserted into the thorax of a mounted mosquito and the solution injected so as to flood the entire insect body. This allowed for circulation of the solution throughout the mosquito, including the JO. In all injection experiments, a ringer solution was injected first as a control. Free fluctuations of the mosquito flagellum were then recorded over the next 10–25 min (depending on the experiment) to observe any potential changes in flagellar mechanics.

**Free fluctuation fitting procedure**. Fast Fourier transforms of the flagellar velocity amplitudes obtained from free fluctuation recordings were calculated for frequencies between 1 Hz and 10 kHz for all mosquito species investigated. Recording measurements below 101 Hz contained a significant level of noise and were excluded from analyses. A forced damped harmonic oscillator function adapted from that used in D. melanogaster[28] was then fitted to flagellar velocity amplitudes between 101 and 1000 Hz. The original function was used for squared displacement amplitudes ($X^2(\omega)$); this modified version was instead used for velocity amplitudes (($\dot{X}(\omega)$) by converting between the function domains:

$$X^2(\omega) = \frac{(F_0/m)^2}{(\omega_0^2 - \omega^2)^2 + \left(\omega \cdot \frac{\omega_0}{Q}\right)^2},$$
(1)

$$X^2(\omega) = (\dot{X}(\omega) \cdot \omega)^2,$$
(2)

where $F_0$ is the external force strength, $m$ is the flagellar apparent mass, $\omega$ is the angular frequency, $\omega_0$ is the natural angular frequency and $Q$ is the quality factor (with $Q = m\omega_0/\gamma$, where $\gamma$ is the damping constant).

The following velocity amplitude fit function was fitted to the data:

$$\dot{X}(\omega) = \frac{F_0/m}{\sqrt{\omega^2 \cdot \left((\omega_0^2 - \omega^2)^2 + \left(\omega \cdot \frac{\omega_0}{Q}\right)^2\right)}}.$$
(3)

This function provided estimates for $F_0/m$, $\omega_0$ and $Q$, which were then utilised to calculate other parameters such as the flagellar best frequency[28]. These data were aggregated across individual mosquitoes, which allowed for population estimates to be made. Further information regarding the fitting procedure is available in the Supplementary Methods. An example of the velocity function fit to free fluctuation data in the active and passive states for a female An. gambiae and a male Cx. quinquefasciatus is given in Supplementary Figure 1a.

**Apparent flagellar mass estimations**. No apparent flagellar mass values have previously been reported for any mosquito species; it was therefore necessary to determine the relevant apparent mass values for the mosquitoes' flagella. We used an adaptation of the procedure previously reported for the Drosophila antennal ear[28]. Individual mosquitoes were passive and free fluctuations of their

passive flagella recorded. The damped harmonic oscillator model described above was then fitted to the resulting velocity spectra.

Assuming that the mosquito auditory system is in a passive state, then mosquito flagellar fluctuations should obey the Equipartition theorem:

$$\frac{1}{2}K\langle x^2 \rangle = \frac{1}{2}k_B T, \tag{4}$$

where $K$ is the effective stiffness of the oscillator, $\langle x^2 \rangle$ is the sum of the squared Fourier displacement amplitudes, $k_B$ is the Boltzmann constant ($1.38 \times 10^{-23}$ J/K) and $T$ is the absolute temperature (estimated at approximately 293 K).

Assuming that $K$ is equal to the spring constant, $K_S$, of the oscillator whilst the mosquito is passive, then the relationship between the spring constant, the apparent flagellar mass, $m$, and the natural frequency, $\omega_0$, of the system can be modified accordingly:

$$K_S = m\omega_0^2, \tag{5}$$

$$m = \frac{k_B T}{\omega_0^2 \langle x^2 \rangle}. \tag{6}$$

Thus, two parameters were required to calculate the apparent flagellar mass: (i) the natural frequency of the passive oscillator and (ii) the sum of the corresponding (fast Fourier transform-derived) squared displacement amplitudes (the receiver's corresponding total fluctuation power, $\langle x^2 \rangle$).

Both of these values were extracted in accordance with published methodologies[28], with the natural frequency being approximated from the velocity amplitude fit function and $\langle x^2 \rangle$ following from:

$$\langle x_i^2 \rangle = \int_0^{\infty} x_i^2(\omega)\,d\omega. \tag{7}$$

It was assumed that the flagellar mass remained constant between active and passive states. For many mosquitoes, passive state fluctuations were recorded before and after pymetrozine exposure and thus a two-state mixed-effects model was used to produce mass estimates (see Statistics section below).

Measurements from 56 Ae. aegypti females (35 before pymetrozine/21 after pymetrozine), 45 Ae. aegypti males (30/15), 50 Cx. quinquefasciatus females (29/21), 54 Cx. quinquefasciatus males (33/21), 50 An. gambiae females (33/ 17) and 31 An. gambiae males (22/9) were included in the final analysis.

**Power gain calculations**. Power gain was estimated by calculating the ratio of the total fluctuation power of the auditory system in its active and passive states (building on a reported D. melanogaster procedure[28]).

The energy content of the passive system is given by the sum of the squared Fourier displacement amplitudes in the passive state, $\langle x_p^2 \rangle$, multiplied by the passive spring constant $k_p$ and a proportionality constant $\alpha$:

$$E_p = \alpha k_p \langle x_p^2 \rangle. \tag{8}$$

The passive spring constant is the product of the apparent flagellar mass, $m$, and the square of the natural best frequency of the system, $\omega_p^2$:

$$k_p = m\omega_p^2. \tag{9}$$

Thus,

$$E_p = \alpha k_p \langle x_p^2 \rangle = \alpha m \omega_p^2 \langle x_p^2 \rangle. \tag{10}$$

Following the same assumptions in the active state provides an equivalent equation:

$$E_\alpha = \alpha k_a \langle x_\alpha^2 \rangle = \alpha m \omega_\alpha^2 \langle x_\alpha^2 \rangle. \tag{11}$$

We defined power gain as:

$$\text{Power gain} = \frac{E_a - E_p}{E_p}. \tag{12}$$

Thus,

$$\text{Power gain} = \frac{\alpha m \omega_a^2 \langle x_a^2 \rangle - \alpha m \omega_p^2 \langle x_p^2 \rangle}{\alpha m \omega_p^2 \langle x_p^2 \rangle}. \tag{13}$$

This equation was reduced to:

$$\text{Power gain} = \frac{\omega_a^2 \langle x_a^2 \rangle}{\omega_p^2 \langle x_p^2 \rangle} - 1. \tag{14}$$

Thus, power gain calculations required estimates of the best frequency in addition to the sum of the squared Fourier displacement amplitudes in both the active and passive (i.e. $CO_2$ sedated) states. Best frequency values were obtained by fitting the damped harmonic oscillator function described above to the fast Fourier transform-derived frequency spectra of the flagellar velocity amplitudes between 101 and 1000 Hz in the active and passive states. The sum of the squared Fourier displacement amplitudes was estimated as in Eq. (7).

Thirty-five Ae. aegypti females, 29 Ae. aegypti males, 28 Cx. quinquefasciatus females 31 Cx. quinquefasciatus males, 33 An. gambiae females and 24 An. gambiae males were included in the final analysis.

**Force-step stimulation recordings**. After mounting a mosquito, a charging electrode was inserted into the mosquito in order to raise its electrostatic potential to −20 V relative to the ground. Two electrostatic actuators were brought into position symmetrically about the flagellum to allow for push and pull electrostatic stimulation of the flagellum. A recording electrode was then inserted at the base of the right pedicel so recordings could be made of compound antennal nerve responses to stimulation. The flagellum was then stimulated using force-step stimuli. Precise measurements of flagellar displacement (via LDV) were recorded in conjunction with simultaneous electrophysiological activity. Supplementary Figure 2a contains examples of the flagellar and antennal nerve responses to a step stimulus for female and male Ae. aegypti.

**Force-step stimulation analysis**. Mosquito apparent flagellar mass estimates were produced as described above. Force-step stimulation analysis then proceeded in accordance with published analyses[25]; for all mosquitoes a two-state model of a single transducer population was utilised. Only displacement data recorded between ±2000 nm for females and ±600 nm for males was included to focus the initial analyses on the most sensitive transducers in each sex and to reduce the potential impact of any further non-auditory nonlinearities.

A single transducer population model was used for fitting rather than a model that could account for two independent populations[33] because no prior studies investigating the existence of multiple independent populations in mosquito species have been reported (in contrast to D. melanogaster, for whom molecularly distinct auditory and non-auditory, also referred to as sensitive and insensitive, populations have been reported[33,51]). Please note that such a single transducer population model can also account for two anatomically opposing transducer populations, which open or close respectively in response to a given antennal displacement. The corresponding mathematical details have been published elsewhere[25].

To account for differences between different species/sexes in the distance between laser focus point and flagellar base (due to differences in flagellar length and anatomy), we normalised the flagellar length of all mosquitoes to a unitary reference point of 1 mm above base (Supplementary Table 1 contains measurements of flagellar length for each sex and species). This normalisation was achieved by multiplying the relevant displacement values by a constant (within an individual mosquito group) factor equivalent to the inverse of the distance between the laser focus point and the base of the flagellum. This normalisation procedure enables direct comparisons between the different mosquito groups.

Twenty-one Ae. aegypti females, 18 Ae. aegypti males, 17 Cx. quinquefasciatus females, 15 Cx. quinquefasciatus males, 18 An. gambiae females and 11 An. gambiae males were included in the final analysis.

**$CAP_{50}$ calculations**. Compound antennal nerve response magnitudes for individual mosquitoes were first normalised to the maximum antennal nerve response value for that individual and were then fitted with a five-parameter saturating sigmoid curve fit:

$$y = y_0 + \frac{a}{\left(1 + e^{\left(\frac{X - X_0}{b}\right)}\right)^c}. \tag{15}$$

The displacement required to produce 50% of the maximum CAP could then be calculated from this curve, with values from individuals within a group being amalgamated to allow for calculation of medians and standard errors. All

mosquitoes included in the force-step stimulation analysis were included in this analysis.

**White noise stimulus experiments**. Male and female mosquitoes were mounted and charged as described above. The force-step stimulation protocol was then utilised to calibrate the maximum flagellar displacement to approximately ±25,000 nm. The protocol was also used to estimate the proportionality coefficient necessary to convert stimulus voltages into force.

A WN stimulus, programmed in PSV 9.1 (Polytec Ltd.), was then provided between 1 and 3200 Hz, with an external attenuation system (Electronics workshop, University of Cologne) enabling stimulus attenuation. A maximum attenuation of 80 dB was applied first, which was then lifted in 5 dB steps until 0 dB was reached. At each step, flagellar fluctuations in response to the stimulation were recorded, with a final, unstimulated (free) fluctuation being taken at the end of the experiment to assess flagellar system health.

The WN stimulus itself was also recorded at each step, which allowed for calculation of the ratio of the flagellar displacement amplitude and stimulus intensity at each frequency and the fitting of a harmonic oscillator model to the resulting data; this enabled calculation of the mechanical sensitivity at each stimulus intensity.

Mechanical sensitivity values for each stimulus level were then fitted using a three-parameter sigmoidal function, with all fits accepted having $R^2$ values ≥0.9. This enabled the estimation of displacement gains by comparing the values for maximum and minimum attenuations obtained from the fit. Supplementary Figure 1c (top) shows an example of such a fit for a *Cx. quinquefasciatus* female.

Seven *Ae. aegypti* females, 7 *Ae. aegypti* males, 13 *Cx. quinquefasciatus* females, 13 *Cx. quinquefasciatus* males, 9 *An. gambiae* females and 7 *An. gambiae* males were included in the final analysis.

**Pure tone stimulus experiments**. Mosquitoes were prepared as above for WN experiments, including utilising the force-step stimulation protocol to estimate the relevant proportionality coefficient for conversion between stimulus voltage and external force. A recording electrode was also inserted into the base of the mosquitoes' JO in order to record antennal nerve responses. Pure tone (sine wave) stimuli were then used to stimulate the antenna. Stimuli covered the range from 15 to 695 Hz in 10 Hz intervals. Mechanical and nerve responses at higher frequencies were found to be negligible compared to the responses within the above frequency range and were thus not included in the analysis.

At every frequency recorded the stimulus lasted continuously for 2.5 s before stopping for a further 2.5 s; this pattern was repeated five times for each frequency tested.

By fitting a sine wave function to a steady-state segment of the displacement response (after having first applied a direct current to remove the flagellar displacement data in order to centre the response on the resting position), an estimate of the peak flagellar displacement at each stimulus frequency was obtained. Applying the same process to the stimulus itself at each frequency tested enabled a ratio of flagellar displacement to stimulus force to be calculated for all frequency values. This sensitivity was calculated for each frequency value and a Gaussian function was fitted to the resulting data in order to estimate maximum and minimum sensitivities (with no assumptions made as to whether the flagellar response was best modelled by the function). Calculating the ratio of maximum and minimum values thus enabled calculation of frequency-dependent displacement gains. Supplementary Figure 1c (bottom) shows an example of the changes in sensitivity as well as the function fit for an *Ae. aegypti* female.

**Mosquito immunohistochemistry**. Following removal of the proboscis, mosquito heads were fixed in 4% paraformaldehyde for 1 h at room temperature[23]. After fixation, heads were embedded in albumin/gelatin, post-fixed in 6% formaldehyde overnight at 4 °C and sectioned (40 µm). Sections were washed in phosphate-buffered saline 0.3% Triton X-100 and blocked in 5% normal goat serum and 2% bovine serum albumin. Primary antibodies used were monoclonal antibody 3C11 (anti-SYNORF1; 1:50; Developmental Studies Hybridoma Bank, University of Iowa, http://dshb.biology.uiowa.edu/) and the conjugated primary antibody anti-HRP-Cy3 (1:500, Jackson ImmunoResearch, Code: 123-165-021). Secondary antibodies used correspond to Alexa Fluor Dyes (1:500; Thermo Fisher). Samples were mounted in DABCO and visualised using a Zeiss 880 confocal microscope.

**Statistical analysis**. Sample sizes for all LDV experiments were determined based on published investigations focussed on Dipteran antennal LDV measurements[28,33,40,62]. Estimates of within-group variation (where appropriate) were calculated as part of the standard statistical tests and were considered reasonable for the field.

Statistical tests for normality (Shapiro–Wilk Normality tests with a significance level of $p < 0.05$) were used for each LDV dataset. These were generally found to be non-normally distributed; thus, median and standard error values are reported throughout.

For the free fluctuation and power gain investigations, ANOVA on ranks tests were used for comparisons between male and female *Ae. aegypti*, *Cx. quinquefasciatus* and *An. gambiae* mosquitoes. For parameters obtained from free

fluctuation fits, ANOVA on ranks tests were used for comparisons within a mosquito group between active, passive and pymetrozine-exposed states. Repeated-measures ANOVA on rank tests were also used to test for differences in free fluctuation fit parameters within a mosquito group between different flagellar states for the ringer, TTX, TeNT and pymetrozine injection experiments. All tests used a significance level of $p < 0.05$.

For the white noise and pure tone stimulation experiments, Mann–Whitney rank-sum tests were used for statistical comparisons between conspecific female and male *Ae. aegypti*, *Cx. quinquefasciatus* and *An. gambiae* mosquitoes with a significance level of $p < 0.05$.

For CAP magnitude experiments, ANOVA on ranks tests were used for comparisons between female *Ae. aegypti*, *Cx. quinquefasciatus* and *An. gambiae* mosquitoes with a significance level of $p < 0.05$.

For the results of the single transducer population fits, ANOVA on ranks tests were used to make comparisons between female and male *Ae. aegypti*, *Cx. quinquefasciatus* and *An. gambiae* mosquitoes with a significance level of $p < 0.05$.

Supplementary Tables 3–17 contain relevant ANOVA values for all statistical comparisons.

The apparent flagellar mass estimates were found to be normally distributed; thus, two-sided paired $t$ tests were used to test for significant differences between before and after pymetrozine exposure states. As no significant differences were found, a two-state mixed-effects model was utilised to account for the fact that not all measurements were independent of each other—this allowed for maximisation of the dataset as well as estimation of mean values.

All statistical testing was done using the SigmaPlot software (Systat Software, Inc.). The two-state mixed-effects model was fitted in R using the lme4 package[69,70].

## Data availability
All data used for analyses in this paper, as well as further details regarding experimental or analytical procedures, are available from the authors.

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

## Acknowledgements

We thank Prof. Jonathan Ashmore for providing the initial TTX stock solution and Shahida Begum for providing all mosquitoes tested. M.P.S. received funding from the Engineering and Physical Sciences Research Council (EP/F500351/1) through UCL CoMPLEX. M.A. received funding through a Marie Sklodowska-Curie Individual Fellowship from the European Commission (H2020-MSCA-IF-2016/752472). N.B.-G. was supported by a UCL Impact PhD studentship (industrial co-sponsor Syngenta). J.S. was supported by a European Research Council grant to J.T.A. (H2020-ERC-2014-CoG/648709/Clock Mechanics). J.T.A. was supported by grants from the Human Frontier Science Program (RGY0070/2011), the Biotechnology and Biological Sciences Research Council (BB/L02084X/1) and the European Research Council (H2020-ERC-2014-CoG/648709/Clock Mechanics).

## Author contributions

M.P.S., N.B.-G., J.S. and J.T.A. contributed to the conception and design of the research. M.P.S. and J.T.A. analysed the data. M.P.S., M.A. and N.B.-G. performed experiments. M.P.S., M.A., J.S. and J.T.A. wrote the manuscript.

## Additional information

**Competing interests:** The authors declare no competing interests.

