## [Peer Review File · Nature Communications]

Reviewers' comments:

Reviewer #1 (Remarks to the Author):

In this manuscript Topping et al. offer an impressive advance in our understanding of mosquito acoustic ecology. There are several exciting and significant results. In addition to for the first time clearly presenting data on several physiological and structural aspects of mosquito hearing organs across species, they propose an exciting mechanism by which males may hear in the swarming environment.

The manuscript is very well written, but in places I think might fail to communicate the relevance of results to a non-specialist audience in places. Therefore, I have focused my comments on improving the clarity of the ms to a more general audience.

Throughout the results section it would be useful to clearly state the question being asked with a particular assay and explicitly connect the data with the answer. There are a few places where it is assumed the reader will know how the result relates to hearing function. For example, the sex and species specific differences in physiology are addressed in the discussion, but it would be very helpful to have some discussion of how these differences might relate to differences in mating biology or hearing function. This is done really well for the SO data and it would be useful to have similar sections for the auditory transducers section.

I believe that all of these assays were conducted on all three species, but there are a few places in the text where it is unclear when you are referring to all males or males of certain species. Also, in figure 5 it looks like only data for *An. gambiae* and *Cx. quinquefasciatus* is being presented. Was this experiment conducted on *Ae. aegypti*?

There are also several results that seem potentially important that are not fully discussed (Some examples: *Cx. quinquefasciatus* females injected significantly more energy than any other species or sex tested, The best flagellar frequency for *An. gambiae* females was higher in the active state than the passive state, *An. gambiae* females produced CAP responses that were significantly smaller than other females and had reduced JO size and innervation). What do these differences mean for differences in the hearing abilities of these species? Do they reflect known differences in their ecology?

Small comments/suggestions:

Pg 2, Line 26- replace "mating negotiations prior to copulation" with "pre-copulatory interactions"

Pg 3, Line 12- Zika is capitalized.

Pg 4, Line 29- I would recommend writing out White Noise and Pure Tone throughout instead of abbreviating them

Figure 1: The resolution of C is very low in my version (such that I can't read the text), just want to check this for publication.

Table 1: I recommend avoiding abbreviation when possible (Q, PYM+, ect.)

Figure 5: I have some suggestions for making this figure more "readable": 1. label the panels in A (Stimulation, Flagellar displacement, Antennal Nerve Response, and Power Spectra of Flagellar displacement) 2. Along the top of A add the difference between the stimulus and 360 Hz SO under the stimulus frequency (for example 345 Hz (-16 Hz)). 3. Add shading or a box around the 355, 365, and 375 Hz columns to highlight where entrainment is found.

Reviewer #2 (Remarks to the Author):

In the manuscript by Topping and colleagues, the authors perform a series of elegant biophysical experiments aimed at characterizing the response properties of 3 species of mosquito 'ears'. The

group are experts at such studies, and have previously carried out similar experiments in other insects (eg., *Drosophila*). The work is rigorous, well executed, and interesting. It extends our foundational understanding of how the ear of a mosquito detects and interprets sounds, most notably, the wing-beat sounds during mating. This manuscript is rich with information and will likely become a reference source for many future studies. I do not have any major concerns, and am enthusiastic about its publication.

Minor comments:

1. p4, lines 5-11. The finding that *Culex* females inject more energy than any other species or sex is very interesting. It would be of interest to a more general audience if the authors could speculate in the discussion section the biological consequence of this finding. Similarly, the authors identify many sex and species specific differences in the biophysical properties of the mosquito ears (Table 1 and 2). Given the authors' expertise in this area, it would be of value to a general audience to discuss what these differences could mean in terms of hearing abilities between sexes of the same species and among different species.
2. p5, line 5; Missing word? "For broadband, WN stimulation the value"
3. Figure 1B. Please make the tickmarks on the X-axis easier to see.
4. The anti-synapsin staining is difficult to visualize on top of the HRP staining. The authors should consider showing the anti-synapsin channel alone, along with the merge.

Reviewer #3 (Remarks to the Author):

The manuscript 'Sex and Species Specific Hearing Mechanisms in Mosquito Flagellar Ears' provides a detailed study of auditory function in three major disease vectors. The authors showed transduction dependent power gain in the ears of all three mosquito species and sex and species differences in the efferent innervation of the Johnston's organ. Quantitative analysis was based on analytical techniques first employed in the analysis of hearing in *Drosophila melanogaster*. To fit important aspects of their data, especially that derived from force-step stimulation they used a two-state, single transducer population model. From the analysis of their data, the authors revealed sex and species-specific variations, including male-specific, highly sensitive transducer populations. In experiments where they systemically blocked neurotransmission, they recorded large amplitude oscillations in male - but not female - flagellar receivers, indicating sexually dimorphic auditory gain control mechanisms.

The objectives of the paper are important and many of the findings are novel and should be of wide interest, not only to those interested in the study of mosquitoes, but to sensory physiologists and neuroscientists in general. The paper is in many ways a tour de force of experimental technique. The paper is difficult to follow in several places and requires either further, or clearer, justification for methods, especially analysis, and conclusions from findings.

General points

Considerations that it is strongly recommended are raised and addressed early in the manuscript.

These are:

1. Why not express the displacements you measure expressed in angular coordinates? This sounds like a trivial request, but it made initial understanding of the data problematic and I was only partially reassured by the compensation you apply (described in the methods) for flagella length differences. Angular coordinates, it is suggested, are unambiguous.
2. The analytical techniques require further and/or justification. This is because, from the geometry of the flagellum prongs and arrangement of the scolopidia in the JO, it would appear that gross mechanical and electrical measurements from the flagellum and JO respectively are the vector summed outputs from two populations of scolopidia with opposing directional mechanosensitivity. Thus applying analytical techniques that have been applied previously for

single cells, or populations of cells with the same functional polarity, does not appear to be appropriate? The summation seems to be made further complex for several reasons including:

- The scolopidia are arranged radially along the prongs of the flagellum. It is suggested, therefore, that measurements of angular stiffness and gross potentials from the JO should take into account the vector sums of all radial populations.
- Do electrophysiological measurements depend on electrode location? It might be thought that signals recorded from near the edge of the JO would be weighted towards particular populations of scolopidia.

Is there anywhere in the paper where these considerations are discussed. It is suggested they have the potential to influence the analysis and interpretation of the results.

Specific points

Page 1, 20-22. For example, male swarm noise does not impair the ability of male mosquitoes to hear female flight tones (Proc Biol Sci. 2018 Jan 31;285(1871). pii: 20171862), and may even enhance their ability (Entomological Review, 2012, 92, 605–621)

Page 2, 4-8. This is certainly a possibility based on your Figure 5A. However, it difficult to translate the level of the sound stimulus in this figure to particle velocity and, indeed, how this would compare with the particle velocity of a female mosquito flight tone measured at the antenna. This is an interesting question still waiting for an answer, which perhaps you can supply?

Page 3, 33- Page 4, 1. Have you assumed that flagella fluctuations are due to fluctuations generated by the individual scolopidia act independently of each other? This is important because, if they influenced each other, then on average, the activities of opposing scolopidia would tend to cancel each other. Would you please comment .

Page 4, 1-4. From an inspection of Figure 1B, this does not appear to be the case for male mosquitoes, apart for narrow frequency bands near their resonance frequency. Passive animals appear to have flagella fluctuation powers significantly (2-3 orders) higher than those of active mosquitoes on the low frequency tails of the tuning curves. Have you an explanation for the apparent increased activity?

P4L7 and throughout: “Wilcoxon signed rank tests” – I’m confused about the choice of this statistical test. Not about using a non-parametric test (which well and clearly justified in the Methods), but rather using a paired test which is designed to test matched samples or related measurements. When testing between e.g. males and females, or other independent samples, Mann-Whitney is the adequate approach. Confusingly, M-W is also called Wilcoxon rank-sum test, so perhaps this is where this confusion arose? Nonetheless, adequate statistical tests should be performed.

P4L10 and throughout: “ANOVA on ranks” – I’m OK with this approach, but I haven’t found the resume tables of these tests (Sources, d.f., F values, p-values). These values are important and should be available at least in supplemental.

P4L2 and Fig 1B: “...flagellar fluctuation powers were significantly higher in the active, metabolically enabled state” – This sentence doesn’t appear to reflect what the data in fig. 1B shows for males. Except in the best frequency, at the lower range the PSD curves of passive states in males are higher than in the active state. I think a clearer, stepwise explanation of what these results are and what they mean would be desirable. Similarly, the sex-differences in the PSD curves weren’t considered.

Page 4, 15-23 Could these differences be due to differences in the geometry of the JOs in different sexes and species that influence, for example, mechanical coupling of individual scolopidia to the prongs and the numbers and distribution of functionally opposing scolopidia?

Page 4, 24-27. Were spontaneous oscillations of male mosquito flagellae affected by pymetrozine?

P5L12: "In all species and both sexes, WN stimulation yielded significantly lower displacement gains than PT stimulation" – I'm confused with this conclusion: weren't WN and PT used to measure different types of gain? Some lines above it was referred that WN was used to calculate broadband, intensity-dependent gains and PT was used to calculate narrowband, frequency-dependent gains. I fail to see then how these 2 approaches are related.

Page 5, 12-18. Perhaps these data could be accounted for if male mosquitoes have a less homogeneous population of scolopodia, sensitive a wider range of frequencies than are found in female mosquitoes (see Lapshin DN, Vorontsov DD (2017) *J. Exp. Biol.* 220: 3927-3938. doi: 10.1242/jeb.152017). Net responses of the scolopodia may be a consequences of homogeneity of frequency selectivity and distribution of scolopodia within the JO, and cancellation though the generation of opposing responses to the same stimulus. It is difficult to assess the significance of your data without having a full understanding of how individual scolopodia contribute to the overall movements of the flagella and the compound electrical responses recorded from the JO.

P5L22: "The receivers of *An. gambiae* females however, showed characteristic intensity-dependent best frequency increases as previously reported for *Drosophila*. Male flagellar best frequencies, in contrast, remained constant up to a distinct force intensity, and then decreased to a new level (Figure S1C)." – This is an interesting and important analysis, and it is somewhat unfortunate to be buried in supplemental. If not here, I hope the authors follow these results in another context.

Page 5, 19-26. Surely, this rather depends on the relationship between the flagellum and its substrate? If the flagellum interacts with the active components (scolopodia) of the substrate through impedance matching, such that physical changes in one component influence the physical properties of another, as with OHCs and the cochlear partition, then it is likely factors such as Q and best frequency of the flagellum vibrations will change.

Page 9, 14-26. It was difficult to understand this data in the context of a single population of scolopodia, as you appear to present it. This issue was raised at the beginning of these comments. Perhaps your presentation could be clearer? Displacement away from the zero position will tend to increase the open probability of MET channels in one population of the scolopodia in the JO and increase the probability of closure of the opposing population. These two probabilities should be different, or there would not be the net changes you observe in the mechanical and voltage responses. Could you please explain how you fit a function apparently designed for MET channels in a single population of scolopodia to data obtained through asymmetric interaction between two opposing populations of scolopodia?

P9: "Auditory transducers mediate sex- and species specific properties in mosquito ears" – It probably reflects more the limitations of this reviewer than the authors, but this Results section would gain with clearer definitions of the main concepts, such transducer gating and gating compliances, and how they are related to the quantified measures. This appeal includes a clarification on how "predicted transducer channel open probabilities" relates to the CAP measures, and relevance of the different stiffness parameters.

Page 10, 4-6. Do you have data for male mosquitoes? The female (and male if available) data appears to be important enough to be in the main text of the paper.

P17: "Nervous system-wide disruption of synaptic, or action potential-dependent, signalling leads to large self-sustained oscillations only in male mosquitoes" – This subsection lacks a basic characterization of the SOs, such as the average frequency and amplitude (and how variable these parameters are). It also lacks a reference on how the onset of the SOs was achieved and controlled.

Page 17, 10-13. As a corollary to this, SOs are blocked by colchicine and temperatures close to 4°C leaving transduction unimpaired (Warren et al., 2010). SOs appear to be a consequence of transduction.

Page 24, L5. Suggested by some for IHCs. Not a widely held view, largely because there are no in vivo measurements yet available to support the suggestion. Please replace 'thought' with 'suggested'.

Page 24, L5. 'inject energy' There appears also another important difference between in situ IHC hair bundle and mosquito flagella motion. This is that energy injection as a consequence of hair bundle motion is attributed to the forward transduction itself. Flagella motion in mosquitoes is due to a mechanism separate from sensory transduction. Forward (sensory) transduction can occur when flagella motion (SOs) is blocked (Warren et al., 2010). Is there more than one source of flagella motion in mosquitoes?

Page 24, 17-20. This appears a too simplistic approach (see also comments above). The forces generated by the radially arranged scolopodia will be vectoral. Scolopodia on other prongs will contribute according to the cosine of the angle of the prongs to the flagella displacement. Contributions will also be influenced by the overall geometry of the JO.

Page 24 28-30. Couldn't Myosin VII A or Dynein be possible candidates?

Page 24, 33-35. Sex and species specific differences could be attributed to differences between species and sexes in JO geometry and relative contribution of sensilla to flagella motion (see also above).

P24L33: Is it possible to advance any behavioural and ecological significance for the "substantial sex and species specific differences"?

Page 25. 2-3. This sentence is not clear. In vertebrates, mechanical feed back is provided by hair bundle (MET) motility (product of forward transduction) and voltage-dependent somatic motility (independent of but dependent on forward transduction). Basis of motility in JO (very likely dynein and ??, coupled to forward transduction?).

Page 25, 7-8. There does not appear to be, anywhere in the paper, a justification for making this assumption, apart from others have used as stated here. As already questioned above, it is not apparent how the model copes with such complexities such as opposing gating springs and the geometry of the JO.

Page 25, 14-21. As already asked above. Could these differences also be accounted, for example, by differences in geometry of the JO and proportions in scolopodial competitive interaction?

P25L18: "The males of all species had transducer modules with (i) a greater total gating spring stiffness, KGS, (ii) larger single channel gating forces, z , and (iii) smaller numbers of predicted transducer channels, N , than conspecific females" – The sex-specific relation between the calculated transducer parameters and the hearing function is an important point of discussion that is lacking; more to the point, how do these parameter values (either taken individually or together) are associated to the higher hearing sensitivity in males?

Page 26, 30-32. Similar behaviour has been observed in the mammalian auditory system, especially in CF bats (J. Neurosci., 2003,23:9508 –9518. See Strogatz SH (1994) Nonlinear dynamics and chaos: with applications to physics, biology, chemistry, and engineering. Reading MA: Addison-Wesley. for a theoretical basis of the observations you report.

Page 27, 7-9. DPs, due to interaction between an external tone and SOs have already been reported (Current Biology 20, 131–136). Difference tones (quadratic) DPs, (the product of a stiffening nonlinearity like that reported in your MS) are recorded near the apical surface of the JO. Cubic distortion, likely to be generated at the level of the neurons, is recorded near the base of the JO (Lapshin, D. N. (2012) Entomol Rev, 92, No. 6, pp. 605–621). Of considerable importance, as mentioned above, is the level of an external tone (female flight tone) required to entrain the SOs.

Minor points

P2L22: "Several studies have proposed potential mechanisms of acoustic signalling between conspecific males and females within these swarms." – Please reference these studies.

P3L1: "Another phenomenon that might offer valuable insights into mosquito acoustic courtship are spontaneously occurring SOs..." – I would suggest that the SOs might first offer insights into the sensory function, well before the mating behaviour. But consider this just a suggestion.

P3L5: Are any references that support that the SOs, and in mosquitoes in particular, are a "pathological signature"?

P3L16: "METs" – This abbreviation wasn't used anywhere in the text.

P4L1 and throughout – The authors favour the term the "passive" in the text and table 1 but use the term "sedated" in figure 1 (and legend). I would suggest using the same term in all instances.

P4L8 and tables: "SE" – Are these values the Standard Errors of the median?

P7L9: "Calculated energy gains" – Shouldn't be power gains?

P9L25: "Female *An. gambiae* produced CAP responses of significantly smaller magnitudes than those seen in females from the two other species" – A statistical test is needed to support this.

P12L11: "Bi-dimensional error bars utilise standard errors." – What error bars?

P13: Table 3 – Please define in the legend all the other parameters.

P14L6: "Paying tribute to their potential roles in modulating mosquito auditory sensitivity, specific focus was placed on the JOs' efferent innervation patterns." – Awkward sentence.

Fig. 4A and legend: The Left/right separation of data in fig 4A is confusing. Why not separate them in 2 different figures?

Figure 5A: It would be useful to identify the different rows with the corresponding quantification: PT Stimulus, flagellar displacement, AN responses, Power spectra.

P25L31: "Conversion of the λ_{90} displacements into angular deflections..." – Since angular deflections facilitates comparisons with other studies, it would be useful to include these values in Table 2.

- I haven't found any reference in the manuscript on if and how the erection of the antennal hairs in *A. gambiae* males was achieved and/or controlled.

Sex and Species Specific Hearing Mechanisms in Mosquito Flagellar Ears

Matthew P. Topping^{1,2,3}, Marta Andrés Miguel^{1,3}, Nicholas Boyd-Gibbins¹, Jason Somers^{1,3} & Joerg T. Albert^{1,2,3,4*}

Point by point list of manuscript revisions suggested by the three reviewers:

Reviewer #1

Reviewer #2

Reviewer #3

Reviewer #1 (Remarks to the Author):

In this manuscript Topping et al. offer an impressive advance in our understanding of mosquito acoustic ecology. There are several exciting and significant results. In addition to for the first time clearly presenting data on several physiological and structural aspects of mosquito hearing organs across species, they propose an exciting mechanism by which males may hear in the swarming environment.

The manuscript is very well written, but in places I think might fail to communicate the relevance of results to a non-specialist audience in places. Therefore, I have focused my comments on improving the clarity of the ms to a more general audience.

Throughout the results section it would be useful to clearly state the question being asked with a particular assay and explicitly connect the data with the answer. There are a few places where it is assumed the reader will know how the result relates to hearing function. For example, the sex and species specific differences in physiology are addressed in the discussion, but it would be very helpful to have some discussion of how these differences might relate to differences in mating biology or hearing function. This is done really well for the SO data and it would be useful to have similar sections for the auditory transducers section.

We thank the reviewer for their suggestions how to improve the clarity of our manuscript by bridging the gap between auditory biophysics and sensory ecology. We fully agree with the necessity of doing this and are particularly grateful for the reviewer's acknowledgement of our efforts to achieve this goal when discussing the potential role(s) of the spontaneous oscillations (SOs) in males. The unique, male-specific phenomenon of SOs lends itself more easily to such interpretation than the more general differences found on the level of auditory transducer function. Encouraged by the reviewer, however, we have made an additional effort to offer some suggestions on the ecological roles, and causes, of the differences found. These are now part of the revised discussion section.

The added paragraphs now state:

“These sex-specific variations match theoretical expectations for transducer populations of different sensitivities⁴⁹ and are also in close agreement with differences found experimentally between sensitive (auditory) and insensitive (wind/gravity) transducers in the *Drosophila* ear³⁰. In mosquitoes, the ears of all males possess more sensitive transducers than conspecific females, suggesting particular ecological specialisations. It seems plausible that the male-specific behaviour of detecting, locating and chasing a female flying by is the ecological context of this transducer variation. Further research is needed to unravel the full extent and functional relevance of sex-specific auditory adaptations in mosquitoes. It is unclear whether specialisation is restricted to particular classes of auditory neurons, such as the most sensitive ones or spiking/non-spiking ones⁴⁸; the greatly diminished CAP amplitudes found in Anopheline females could hint at a specific reduction of spiking neurons. The functional investigation of these extensive sexual dimorphisms, however, has just started.”

“From a sensory ecological perspective, it seems noteworthy here that bidirectional acoustic interactions have been reported between males and females flying in couple¹⁰, perhaps hinting at a female choice component in mosquito mating³⁹. Also, roles of audition beyond mating might include predator avoidance (in both males and females)⁴⁰ or host finding (in females only). Corresponding phonotactic responses have indeed been reported for *Culex* sp. females⁴¹, possibly explaining why *Culex quinquefasciatus* was the only species where female baseline auditory amplification exceeded that of males.”

I believe that all of these assays were conducted on all three species, but there are a few places in the text where it is unclear when you are referring to all males or males of certain species. Also, in figure 5 it looks like only data for *An. gambiae* and *Cx. quinquefasciatus* is being presented. Was this experiment conducted on *Ae. aegypti*?

It is true that we made an effort to conduct all experiments in both sexes of all species, wherever this was possible. In the case of the SO entrainment experiments presented in Figure 5, we were not able to conduct these in *Aedes aegypti*. For those experiments we depended on the spontaneous occurrence of SOs in the ears of males. This was happening frequently in males of *Cx. quinquefasciatus* and *An. gambiae*. For reasons beyond our control (or understanding), males of *Ae. aegypti* did not display spontaneous SOs sufficiently often in our experimental setup to enable these experiments. We are sorry for not having made this sufficiently clear and have reworded the respective legend (Figure 5B).

It now states:

“There is no data for *Ae. aegypti* males as they did not show spontaneous SOs under our experimental conditions.”

There are also several results that seem potentially important that are not fully discussed (Some examples: *Cx. quinquefasciatus* females injected significantly more energy than any other species or sex tested, The best flagellar frequency for *An. gambiae* females was higher in the active state than the passive state, *An. gambiae* females produced CAP responses that were significantly smaller than other females and had reduced JO size and innervation). What do these differences mean for differences in the hearing abilities of these species? Do they reflect known differences in their ecology?

The ecological explanation of the observed sexual dimorphisms in auditory function is the ultimate goal of our efforts. At present, however, we can at best make some educated and plausible suggestions (as we tried for the SOs). In the cases of the other questions, such as the reduced size of the JO in Anopheles females or their over proportionately reduced CAP amplitudes, we are left to speculation. In the light of previous published reports, we have added a suggestion that much of the observed sexual dimorphism may be restricted to specific classes or types of JO neurons, which differ in their sensory ecological roles. In our compound analyses (both electrophysiological and mechanical ones) we cannot distinguish between those classes easily and we have made a conscious effort not to do this here in order to stay on scientifically 'safe ground'. However, we have amended the discussion section according to the reviewer's enquiry (see also answer above).

We added (as explained above):

"Further research is needed to unravel the full extent and functional relevance of sex-specific auditory adaptations in mosquitoes. It is unclear whether specialisation is restricted to particular classes of auditory neurons, such as the most sensitive ones or spiking/non-spiking ones⁴⁸; the greatly diminished CAP amplitudes found in Anopheline females could hint at a specific reduction of spiking neurons. The functional investigation of these extensive sexual dimorphisms, however, has just started."

Small comments/suggestions:

Pg 2, Line 26- replace "mating negotiations prior to copulation" with "pre-copulatory interactions"

Thanks for pointing this out. Text amended accordingly.

Pg 3, Line 12- Zika is capitalized.

Thanks for pointing this out. Text amended accordingly.

Pg 4, Line 29- I would recommend writing out White Noise and Pure Tone throughout instead of abbreviating them

Thanks for pointing this out. Text amended accordingly.

Figure 1: The resolution of C is very low in my version (such that I can't read the text), just want to check this for publication.

Thanks for pointing this out. The final images will be of publication quality resolution.

Table 1: I recommend avoiding abbreviation when possible (Q, PYM+, ect.)

Thanks for pointing this out. We have reduced the use of abbreviations throughout the manuscript.

Figure 5: I have some suggestions for making this figure more "readable": 1. label the panels in A (Stimulation, Flagellar displacement, Antennal Nerve Response, and Power Spectra of Flagellar displacement) 2. Along the top of A add the difference between the stimulus and 360 Hz SO under the stimulus frequency (for example 345 Hz (-16 Hz).). 3. Add shading or a box around the 355, 365, and 375 Hz columns to highlight where entrainment is found.

Thanks for suggesting this. Figure 5 has been amended accordingly.

Reviewer #2 (Remarks to the Author):

In the manuscript by Topping and colleagues, the authors perform a series of elegant biophysical experiments aimed at characterizing the response properties of 3 species of mosquito 'ears'. The group are experts at such studies, and have previously carried out similar experiments in other insects (eg., *Drosophila*). The work is rigorous, well executed, and interesting. It extends our foundational understanding of how the ear of a mosquito detects and interprets sounds, most notably, the wing-beat sounds during mating. This manuscript is rich with information and will likely become a reference source for many future studies. I do not have any major concerns, and am enthusiastic about its publication.

Minor comments:

1. p4, lines 5-11. The finding that *Culex* females inject more energy than any other species or sex is very interesting. It would be of interest to a more general audience if the authors could speculate in the discussion section the biological consequence of this finding. Similarly, the authors identify many sex and species specific differences in the biophysical properties of the mosquito ears (Table 1 and 2). Given the authors' expertise in this area, it would be of value to a general audience to discuss what these differences could mean in terms of hearing abilities between sexes of the same species and among different species.

This point goes, of course, to the core of our study and we thank the reviewer for raising it!

Our study offers multiple new insights into auditory modifications between mosquito species or males and females. Only very few of those can, at this stage, be safely linked to a specific ecological purpose. Appreciating the importance of the question, however, we have made changes to the manuscript text, which offer some hypotheses (see also responses to reviewer 1): In the discussion we, e.g. now write:

"Also, roles of audition beyond mating might include predator avoidance (in both males and females)⁴⁰ or host finding (in females only). Corresponding phonotactic responses have indeed been reported for *Culex sp.* females⁴¹, possibly explaining why *Culex quinquefasciatus* was the only species where female baseline auditory amplification exceeded that of males."

2. p5, line 5; Missing word? "For broadband, WN stimulation the value"

Thanks for pointing this out. Text amended accordingly.

3. Figure 1B. Please make the tickmarks on the X-axis easier to see.

Thanks for pointing this out. Figure has been amended accordingly.

4. The anti-synapsin staining is difficult to visualize on top of the HRP staining. The authors should consider showing the anti-synapsin channel alone, along with the merge.

Thanks for pointing this out. The Figure, which is central to our paper, has been amended accordingly and a new version with the individual channels is presented in the supplement (Figure S5).

Reviewer #3 (Remarks to the Author):

The manuscript 'Sex and Species Specific Hearing Mechanisms in Mosquito Flagellar Ears' provides a detailed study of auditory function in three major disease vectors. The authors showed transduction dependent power gain in the ears of all three mosquito species and sex and species differences in the efferent innervation of the Johnston's organ. Quantitative analysis was based on analytical techniques first employed in the analysis of hearing in *Drosophila melanogaster*. To fit important aspects of their data, especially that derived from force-step stimulation they used a two-state, single transducer population model. From the analysis of their data, the authors revealed sex and species-specific variations, including male-specific, highly sensitive transducer populations. In experiments where they systemically blocked neurotransmission, they recorded large amplitude oscillations in male - but not female - flagellar receivers, indicating sexually dimorphic auditory gain control mechanisms.

The objectives of the paper are important and many of the findings are novel and should be of wide interest, not only to those interested in the study of mosquitoes, but to sensory physiologists and neuroscientists in general. The paper is in many ways a tour de force of experimental technique. The paper is difficult to follow in several places and requires either further, or clearer, justification for methods, especially analysis, and conclusions from findings.

General points

Considerations that it is strongly recommended are raised and addressed early in the manuscript. These are:

1. Why not express the displacements you measure expressed in angular coordinates? This sounds like a trivial request, but it made initial understanding of the data problematic and I was only partially reassured by the compensation you apply (described in the methods) for flagella length differences. Angular coordinates, it is suggested, are unambiguous.

We are very grateful to the reviewer for pointing this out and the request is by no means trivial. We fully agree with the reviewer that an angular axis is in a number of ways preferable, because of its facilitation of easier comparisons between species and sexes. In an initial draft of the paper we even had prepared a rotational stiffness plot but we then decided against it simply to reduce the paper's 'technical bulkiness'. We have now added a rotational stiffness comparison to the supplement.

The reasoning behind our original decision to choose displacement based compliance plots was to enable the use of the original simple gating spring model, which was formulated for length and not angular units. In order to still allow for comparisons between mosquitoes we have, however, normalised the parameter values to a unitary flagellum reference point of 1 mm above the base. Now we present both length (Figure 2) and angular units (Figure S3) in the paper. Thanks again for making this suggestion.

2. The analytical techniques require further and/or justification. This is because, from the geometry of the flagellum prongs and arrangement of the scolopidia in the JO, it would

appear that gross mechanical and electrical measurements from the flagellum and JO respectively are the vector summed outputs from two populations of scolopidia with opposing directional mechanosensitivity. Thus applying analytical techniques that have been applied previously for single cells, or populations of cells with the same functional polarity, does not appear to be appropriate?

We thank the reviewer to raise this important point and for giving us an opportunity here to explain our procedure in more detail. The simple, single population gating spring model we applied here, was developed for the *Drosophila* antennal ear [REF: Albert JT, Nadrowski B, Göpfert MC. Mechanical signatures of transducer gating in the *Drosophila* ear. *Curr Biol.* 2007;17(11):1000-6.]. The fly's ear is comprised by two anatomically opposing populations of neurons, very much like we assumed for two opposing prongs in the mosquito ear. The gating spring model takes this into account. In the supplement of the above cited paper, we explain the situation in detail. The respective paragraph in the supplement of the Albert et al (2007) paper starts with:

“The hearing organ of *Drosophila* comprises two populations of sensory neurons that perpendicularly connect to the receiver's anterior and posterior sides and are likely to be alternately stretched and compressed as the receiver moves back and forth [S3]. Such system may be described by a gating-spring model with two opposed populations of gating-spring modules, the open probabilities of which are inversed: the respective open probabilities of the two populations are $p_o(X)$ and $p_o(2X)$, which means that when one channel population is mostly in the open state, the channels of the other population will mostly be closed.”

We show that in such a case the simple gating spring model, which assumes mirror symmetric (but otherwise identical) transducers in the two opposing neuronal populations, can describe the transducer mechanics. Beyond the mathematical illustration we present in the above paper, the ‘physical’ reason for this is simply given by the fact that in terms of gating compliances, the opening or closing of ion channels are equivalent acts! In both cases, the system's effective stiffness will drop if the ion gate swings. So the (opposing and mirror symmetric) compliances of both populations simply superimpose and add to each other, this is reflected in the model we use. We apologise for not having made this point clearer. We have added a sentence to the materials and methods part.

It now reads:

“Please note that such a single transducer population model can also account for two anatomically opposing transducer populations, which open or close respectively in response to a given antennal displacement. The corresponding mathematical details have been published elsewhere [22].”

The summation seems to be made further complex for several reasons including:

a. The scolopidia are arranged radially along the prongs of the flagellum. It is suggested, therefore, that measurements of angular stiffness and gross potentials from the JO should take into account the vector sums of all radial populations.

This is an interesting aspect of the function of the mosquito JO. It is relevant indeed for some of the quantitative assessments we carried out, the most important parameters here being

the energy estimates per JO neuron, the number of predicted transducers per neuron or the nerve output per neuron.

As the reviewer correctly points out, the question boils down to how many neurons are activated if the mosquito flagellum is moved within a single plane, as done in our study. The current assumption in the field is that the up to ~16,000 neurons in the mosquito JO are divided into individual angular units, arranged and compartmentalised by the prongs. This arrangement would be highly beneficial for the ear's role as sound localiser as the pair of opposing prongs activated would already clearly indicate the plane that the sound source must lie in. The remaining (180 degrees) disambiguation could be made by using binaural input.

In this scenario, the angular resolution of sound source localisation would be given by the angular separation of prongs. This is ~5 degrees in males. It is then possible to estimate the effective number of activated neurons in a simplified 'vectorial spread' scenario, which assumes that each pair of opposing prong is stimulated according to their vectorial participation in the plane of stimulation (thus being fully stimulated within the stimulus plane and not stimulated, at all, if perpendicularly to that plane). That would mean that in each plane of stimulation the number of participating neurons (activated or deactivated) would be almost 1/2 to 2/3 of all neurons (~10,000 neurons in males).

This would then imply that the energy injection per mosquito neuron would be 20-times weaker than in *Drosophila* and that the number of predicted transducer channels per neuron were also likely to be at least one order of magnitude smaller, although both use the same type of sensory cell, namely chordotonal neurons. While not impossible, this scenario is highly unlikely, after also taking into account that a vectorial spread would make sound localisation much more difficult. However, if we assume that the prongs form more or less independent functional units then we end up with very similar numbers as those reported for the fruit fly.

We think that this is rather strong circumstantial evidence that our basic assumption is correct: the neurons attached to individual prongs act more or less separately from each other. This scenario could also help to explain why females have similar baseline energies as males even though they have less neurons. Females are reported to have fewer prongs than males, meaning that they may have similar numbers of neurons per prong. The major difference between the sexes is the higher angular resolution of the male ear here, not the enhanced baseline energy injection within one plane.

We have added some of these considerations to the respective parts of our manuscript. This has indeed added to the sensory ecological interpretation of our results. We thank the reviewer for guiding us into this direction.

The new part of the results section now reads:

"As mentioned previously, however, the neurons of the mosquito JO are grouped in prongs. Prongs are radially arranged cuticular processes, to which numbers of neurons are attached. This arrangement is thought to be the structural basis for the mosquitoes' exquisite ability to localise a sound source. Male JOs possess ~70 prongs, which would, based on purely structural considerations, correspond to a ~5° angular resolution¹⁴. One particular question that has remained unclear is the degree of mechanical separation between neighbouring

prongs. In other words: If the flagellum is displaced within one plane, does the excitation spread across multiple prongs or does it remain restricted to the prongs within the plane of flagellar displacement? Here, our data can at least provide first circumstantial evidence indicating that the prongs appear to be mechanically largely separated from each other.

If there was a vectorial spread of neuronal excitation across the various prongs of the male JO, then the proportion of effectively responding neurons would be >50% for each plane of stimulation. This would not only blur spatial resolution and impair sound source localisation but also imply that the energy contributions per neuron would be at least ~20-times lower than those of *Drosophila*. If however one assumes mechanical separation then our data would represent contributions from neurons between two anatomically opposing prongs only.

There are ~16,000 neurons divided into ~70 prongs in the male *Ae. aegypti* JO¹⁹. The power gain values of this study would thus reflect the contributions of ~460 neurons (in two opposing prongs). In *Drosophila* (due to differences in functional anatomy) all JO neurons (~480 in total³⁸) are likely to contribute; thus the total number of contributing neurons would be roughly the same, explaining the almost identical levels of power gain. This may also indicate that the levels of baseline energy injection are a conserved feature across the scolopidia of Dipteran insects.

The extent of energy injection between male and female mosquitoes was broadly similar across all three species tested, although neuronal numbers are reported to differ by a factor of ~2. Again, the fact that the neurons in the female JO are arranged into fewer prongs is likely to contribute to the equal levels of male and female power gain. These relations may reflect an evolutionary trade-off sacrificing angular resolution for absolute sensitivity.”

b. Do electrophysiological measurements depend on electrode location? It might be thought that signals recorded from near the edge of the JO would be weighted towards particular populations of scolopidia.

Extracellular recordings, such as CAP recordings or other local field potentials (LFP) do indeed vary with the location of the electrode and the distance to the source of the potentials. From our long standing work with LFP recordings in the fruit fly, we are very much aware of these conditions and did not conclude anything which would be distorted by these relations. Two major things are here of importance:

- 1) Our mechanical and electrical recordings were here deliberately restricted to the most sensitive population of neurons, both in males and females. These are characterised by being the first saturating nonlinearity in both CAP recordings and gating compliances. The relative contributions of individual neurons within that group is not relevant here. Future analysis will have to address these highly important and interesting questions and analyse individual neurons or individual subpopulations. Our study, however, made an explicit effort to steer clear of these questions.
- 2) We made a great effort to average a large number of individual CAP recordings, which act to partly average out electrode position effects. Again, we did not draw any conclusion nor extract any parameters which would crucially depend on this question.

Thanks again for raising these issues, they clearly point to the plethora of questions that remain to be asked and answered.

Is there anywhere in the paper where these considerations are discussed. It is suggested they have the potential to influence the analysis and interpretation of the results.

We have now added new paragraphs to address the above issues around our modelling and the possible role of prongs.

Specific points

Page 1, 20-22. For example, male swarm noise does not impair the ability of male mosquitoes to hear female flight tones (Proc Biol Sci. 2018 Jan 31;285(1871). pii: 20171862), and may even enhance their ability (Entomological Review, 2012, 92, 605–621)

Thanks for highlighting this. It is important to bear in mind, however, that our knowledge about the acoustic landscape of mosquito swarms, and the possible noise produced therein, is still very limited.

We have amended the respective introduction paragraph, it now reads:

“Swarms thus form part of the mosquitoes’ natural acoustic space and their corresponding signal-to-noise ratios, as well as resulting amplification and filtering challenges, can be expected to be vastly different for male and female ears. Several studies have proposed potential mechanisms of acoustic signalling between conspecific males and females^{10, 11, 12, 13, 17, 18} but few have discussed these within the context of flying animals^{19, 20} or related these to the specific environment of the swarm¹⁹.”

We have also added the below sentences to the discussion:

“SOs might be part of an enhanced sensing landscape, as has been proposed as an emergent property of mobile animal groups such as mosquito swarms [REF: Berdahl A, Torney CJ, Ioannou CC, Faria JJ, Couzin ID. Emergent Sensing of Complex Environments by Mobile Animal Groups. Science. 2013;339(6119):574-6.]. Indeed it has been suggested that a male mosquito’s own wingbeat is a vital constituent of signal detection in his auditory system¹⁹.”

Page 2, 4-8. This is certainly a possibility based on your Figure 5A. However, it difficult to translate the level of the sound stimulus in this figure to particle velocity and, indeed, how this would compare with the particle velocity of a female mosquito flight tone measured at the antenna. This is an interesting question still waiting for an answer, which perhaps you can supply?

We couldn’t agree more with the reviewer’s intrigue about the question of SO entrainment and its intensity dependences. It would, however, be beyond the scope of this already very dense biophysical assessment to explore this question directly. We will follow up on this suggestion in a future study, for which we have here tried to lay the ground. We are sorry not to be able to answer the question more directly.

Page 3, 33- Page 4, 1. Have you assumed that flagella fluctuations are due to fluctuations generated by the individual scolopodia act independently of each other? This is important because, if they influenced each other, then on average, the activities of opposing scolopodia would tend to cancel each other. Would you please comment .

We thank the reviewer for giving us the opportunity to clarify this important point. Please see the answer above (Reviewer 3, point 2.), where we explain these relations in more detail and also say how we addressed this in the revised manuscript. In short, neither with regard to CAPs nor with regard to gating compliances will the effects of opposing scolopodia cancel

each other out. This is not theoretically expected from our model nor experimentally observed in our recordings.

Page 4, 1-4. From an inspection of Figure 1B, this does not appear to be the case for male mosquitoes, apart for narrow frequency bands near their resonance frequency. Passive animals appear to have flagella fluctuation powers significantly (2-3 orders) higher than those of active mosquitoes on the low frequency tails of the tuning curves. Have you an explanation for the apparent increased activity?

Thanks for raising this point, this is an unfortunate impression given by a lack of clarity in our description of how we produced the fits and estimated the fluctuation powers. The sentence we wrote is true for males and females. Due to the fact that we used the simplest version of modelling available, we also only used a single harmonic oscillator model to fit the fluctuations. In males, this leads to a slightly poorer capture of the lower frequency tail in active, but quiescent male flagella. In terms of the calculated power gain, however, this differences are not relevant for any of our conclusions. We have now added a supplemental figure image (Figure S1C) that shows how we did the fits and how this leads to some systematic deviations, which however have a maximum error <25%, typically much less. Also, we applied this paradigm across all animals and both sexes, which further reduces the errors of distortions. Finally, doing so enables the direct comparisons to previously published *Drosophila* data, which are important. Future follow up analyses can conduct a methodological comparison between different approaches to estimate the power gains for *Drosophila* and mosquitoes. This will require repeating experiments in *Drosophila* as well and was therefore far beyond of what we can achieve here. Again, we are aware of these residual discrepancies and did not draw any conclusions which would be affected by these.

P4L7 and throughout: “Wilcoxon signed rank tests” – I’m confused about the choice of this statistical test. Not about using a non-parametric test (which well and clearly justified in the Methods), but rather using a paired test which is designed to test matched samples or related measurements. When testing between e.g. males and females, or other independent samples, Mann-Whitney is the adequate approach. Confusingly, M-W is also called Wilcoxon rank-sum test, so perhaps this is where this confusion arose? Nonetheless, adequate statistical tests should be performed.

Thanks for highlighting this. We have changed the respective parts.

P4L10 and throughout: “ANOVA on ranks” – I’m OK with this approach, but I haven’t found the resume tables of these tests (Sources, d.f., F values, p-values). These values are important and should be available at least in supplemental.

Thanks for highlighting this. We have added the ANOVA tables to the supplement.

P4L2 and Fig 1B: “...flagellar fluctuation powers were significantly higher in the active, metabolically enabled state” – This sentence doesn’t appear to reflect what the data in fig. 1B shows for males. Except in the best frequency, at the lower range the PSD curves of passive states in males are higher than in the active state. I think a clearer, stepwise explanation of what these results are and what they mean would be desirable. Similarly, the sex-differences in the PSD curves weren’t considered.

The fact that fluctuation powers were higher in all active states as compared to the passive ones holds fully true. The superficial impression from Figure 1B relies on the above explained systematic mismatch between fit and data, especially visible for quiescent male antennae. Even with these mismatches however, the resulting powers are significantly higher in all active, quiescent antennae as compared to their passive states. It is very difficult

to extract the energies from the visual inspection of the plots alone and we understand that this can sometimes be misleading. We hope that the new supplemental figure helps to explain some of these unfortunate superficial impressions. In summary, all active states were of significantly higher energy gains than all passive states. The smallest ratio between active and passive states (*Cx. quinquefasciatus* males) was here >250% (in terms of absolute mechanical energy), whilst the error from the fit mismatch (see above) is ~25%, so we can safely conclude that all active states are more energetic than all passive states.

Page 4, 15-23 Could these differences be due to differences in the geometry of the JOs in different sexes and species that influence, for example, mechanical coupling of individual scolopodia to the prongs and the numbers and distribution of functionally opposing scolopodia?

The geometry of force transmission to the transducers is indeed a possible contributor to some of the differences between sexes and species. We are very grateful for this suggestion and have also added a paragraph to the manuscript. Most remarkable about our findings, however, is how the differences between male and female transducers (and also between species) phenocopy the differences previously reported for different populations of transducers in *Drosophila* (where they have been shown to correspond to molecular differences in the build-up of the transducer machineries) and also how they correspond to theoretical predictions previously made.

The new discussion paragraph reads:

“These sex-specific variations match theoretical expectations for transducer populations of different sensitivities⁵⁴ and are also in close agreement with differences found experimentally between sensitive (auditory) and insensitive (wind/gravity) transducers in the *Drosophila* ear, where they have also been linked to a differential molecular make-up³³. In addition to possible molecular specialisations, variations in transducer geometry (which modify force transmission between the antennal receiver and different JO cilia) could further contribute to the differences observed in both *Drosophila* and mosquitoes.”

Page 4, 24-27. Were spontaneous oscillations of male mosquito flagellae affected by pymetrozine?

Pymetrozine abolished all SOs (both spontaneous and pharmacologically induced).

Please see also our previous paragraph:

“Male flagellar receivers from all species showed the same behaviour in response to both TTX and TeNT injections: large-amplitude SOs (Figure 4A, right; Figure 4B, right) which closely resembled spontaneous SOs. In each case, the frequencies of the pharmacologically-induced SOs were lower than the flagellar best frequencies of the ringer-injected control state (Figure 4B, right). Subsequent injection of the transduction-blocker pymetrozine abolished SOs in all cases (Figure 4A, right).”

P5L12: “In all species and both sexes, WN stimulation yielded significantly lower displacement gains than PT stimulation” – I’m confused with this conclusion: weren’t WN and PT used to measure different types of gain? Some lines above it was referred that WN was used to calculate broadband, intensity-dependent gains and PT was used to calculate narrowband, frequency-dependent gains. I fail to see then how these 2 approaches are related.

We are sorry for not having been clear here. The gains we describe are all displacement gains.

Displacement gain can be measured as the ratio between the maximum and minimum displacement response along a continuous variation of one stimulus parameter. For example, when applying WN we varied the stimulus intensity only and then determined the peak displacement sensitivity (displacement divided by intensity) for each intensity. We then plotted these individual displacement sensitivities against the intensities, which rendered a sigmoid curve (see Figure S2A, top), which allowed for calculating the ratio between the two tail ends. This is the WN displacement gain we talk about. This procedure gives an answer to the question: How does the response sensitivity improve by changing stimulus intensity (across all frequencies, therefore using a broadband stimulus)?

For PT stimulation we picked a medium-range intensity and played this to the antenna at different frequencies. Plotting the resulting flagellar displacements against the individual frequencies, gives a frequency response function. This curve also allows for calculating a displacement gain, namely by fitting a modified Gaussian (see Figure S2A, bottom) and comparing the ratio between the maximum and minimum values. This procedure gives an answer to the question: How does the response sensitivity improve by changing stimulus frequency (at a constant, medium-range stimulus intensity)?

In summary, our results indicate that the system response displays intensity dependence, reflecting a compressive nonlinearity, but that its compression (the ratio of our displacement gains) is considerably lower (by at least a factor of 2.5) than in the case of frequency dependence. In other words, for mosquitoes it seems that even for loud signals, the need for getting the frequency right is very high!

Page 5, 12-18. Perhaps these data could be accounted for if male mosquitoes have a less homogeneous population of scolopidia, sensitive a wider range of frequencies than are found in female mosquitoes (see Lapshin DN, Vorontsov DD (2017) J. Exp. Biol. 220: 3927-3938. doi: 10.1242/jeb.152017). Net responses of the scolopidia may be a consequences of homogeneity of frequency selectivity and distribution of scolopidia within the JO, and cancellation though the generation of opposing responses to the same stimulus. It is difficult to assess the significance of your data without having a full understanding of how individual scolopidia contribute to the overall movements of the flagella and the compound electrical responses recorded from the JO.

The possible existence of different populations of neurons or transducers is an intriguing one, indeed. We agree with the reviewer that the relative contributions of different populations of neurons are likely to have contributed to our data, and the differences we observe. Our measurements and models are all adapted to compound responses, both mechanically (compliances) and electrically (CAPs). We are fully aware of the construction and limitations of our models, which we have long used in, and optimised from, our work in *Drosophila*. The power of these models is that they can provide highly valuable insights even without having a full understanding of individual populations, or even scolopidia. The ambition is, of course, to achieve this individual understanding for all 16,000 neurons in the male JO, but on the road to this aim compound approaches can help to guide the way. None of our conclusions violate the underlying limitations but we completely agree with the reviewer that the eventual goal should be to understand the mechanistic on single cell and single molecule level! The suggested reference, which we have already cited, further supports this point.

P5L22: “The receivers of *An. gambiae* females however, showed characteristic intensity-dependent best frequency increases as previously reported for *Drosophila*. Male flagellar best frequencies, in contrast, remained constant up to a distinct force intensity, and then decreased to a new level (Figure S1C).” – This is an interesting and important analysis, and it is somewhat unfortunate to be buried in supplemental. If not here, I hope the authors follow these results in another context.

We agree wholeheartedly that the complex forms of intensity dependent nonlinearities/compressive nonlinearities we observed are intriguing from a biophysical point of view and highly likely to carry greater ecological relevance. This is the reason why we definitely would like to present this data here, which we feel is the appropriate context. We were also conscious, however, of the already dense body of text, so a detailed follow up will have to take place in another context. We would like to reassure the reviewer that we will make an effort in this regard!

Page 5, 19-26. Surely, this rather depends on the relationship between the flagellum and its substrate? If the flagellum interacts with the active components (scolopodia) of the substrate through impedance matching, such that physical changes in one component influence the physical properties of another, as with OHCs and the cochlear partition, then it is likely factors such as Q and best frequency of the flagellum vibrations will change.

This is fully correct. The very idea of inherent force reciprocity between the two substrates, the *passive* flagellum and the *active* scolopodia, means that the functions and properties of one can be assessed, or predicted, by studying the respective other. For example, the opening of ion channels can be predicted by the gating compliances and the properties of active transducer modules will change the mechanics of the external receiver. These are the bases of changes of Q, power gain or best frequency from hair cells to insect antenna. We hope to have added a valuable new example here.

Page 9, 14-26. It was difficult to understand this data in the context of a single population of scolopodia, as you appear to present it. This issue was raised at the beginning of these comments. Perhaps your presentation could be clearer? Displacement away from the zero position will tend to increase the open probability of MET channels in one population of the scolopodia in the JO and increase the probability of closure of the opposing population. These two probabilities should be different, or there would not be the net changes you observe in the mechanical and voltage responses. Could you please explain how you fit a function apparently designed for MET channels in a single population of scolopodia to data obtained through asymmetric interaction between two opposing populations of scolopodia?

Thanks again for asking this question. For a detailed answer please see our previous commentary. The detailed mathematical explanation can also be found in the supplement of Albert et al (2007) “Mechanical Signatures of Transducer Gating in the *Drosophila* Ear” in *Current Biology*. The original model was already devised for two opposing populations.

P9: “Auditory transducers mediate sex- and species specific properties in mosquito ears” – It probably reflects more the limitations of this reviewer than the authors, but this Results section would gain with clearer definitions of the main concepts, such transducer gating and gating compliances, and how they are related to the quantified measures. This appeal includes a clarification on how “predicted transducer channel open probabilities” relates to the CAP measures, and relevance of the different stiffness parameters.

We very much appreciate the reviewer’s suggestion to provide more conceptual clarity here. We have made changes to the Results, Discussion and Methods sections to address this. We have also added a new reference that explains the theoretical fundamentals of our

approach [REF: Hudspeth AJ, Choe Y, Mehta AD, Martin P. Putting ion channels to work: Mechanoelectrical transduction, adaptation, and amplification by hair cells. *Proc Natl Acad Sci U S A* **97**, 11765-11772 (2000).].

Page 10, 4-6. Do you have data for male mosquitoes? The female (and male if available) data appears to be important enough to be in the main text of the paper.

We have added the requested data to the supplement (Figure S4).

P17: “Nervous system-wide disruption of synaptic, or action potential-dependent, signalling leads to large self-sustained oscillations only in male mosquitoes” – This subsection lacks a basic characterization of the SOs, such as the average frequency and amplitude (and how variable these parameters are). It also lacks a reference on how the onset of the SOs was achieved and controlled.

We thank the reviewer for pointing out this lack of information. We have now added a table to the supplement (Table S2), which details and compares the two types of SOs; the spontaneously occurring ones, which we did not induce or control, and the pharmacologically induced ones, which followed from our TTX or TeNT injections. The injection protocol we used to induce the SOs is detailed in the methods section.

Page 17, 10-13. As a corollary to this, SOs are blocked by colchicine and temperatures close to 4°C leaving transduction unimpaired (Warren et al., 2010). SOs appear to be a consequence of transduction.

The interrelation between SOs and transduction is indeed a mechanistically important one. We assume that reviewer meant to say “SOs appear not to be a consequence of transduction” instead of “SOs appear to be a consequence of transduction” as the cited paper reports on some residual transduction in mosquitoes injected with the microtubule polymerization inhibitor colchicine but a loss of SOs in these animals.

The interrelation between microtubular cytoskeleton, mechanotransducer machinery and molecular motors is at the core of active transducer modules. At present no study has shown an independence of SOs and transduction. Warren et al. 2010 (“The dynein-tubulin motor powers active oscillations and amplification in the hearing organ of the mosquito”) found a reduction of the CAPs to 1/3 of the previous amplitude (as judged by Figure 5e from that publication) after microtubule disruption; thus transduction was probably substantially impaired. *Drosophila* studies have found that genetic disruption of the microtubular apparatus does partly abolish transduction (Bechstedt et al., “A doublecortin containing microtubule-associated protein is implicated in mechanotransduction in *Drosophila* sensory cilia”, 2010).

In all our experiments with mosquitoes and fruit flies, we found that mutations that affect the microtubular machinery affect both transduction and amplification (or SOs). Also, application of the nanchung/inactive agonist pymetrozine abolishes transduction and SOs. It has to be left to future studies to resolve this question; for now we can only state that both phenomena are closely interlinked.

Page 24, L5. Suggested by some for IHCs. Not a widely held view, largely because there are no in vivo measurements yet available to support the suggestion. Please replace 'thought' with 'suggested'.

We thank the reviewer for this comment, which accurately describes the current view on amplification and energy injection for mammalian inner hair cells (IHCs). We are, however, not referring to the mammalian cochlea here but more generally to hair cells of the vertebrate inner ear (including bullfrogs etc.), which are commonly referred to as inner-ear hair cells (as opposed to IHCs).

Page 24, L5. 'inject energy' There appears also another important difference between in situ IHC hair bundle and mosquito flagella motion. This is that energy injection as a consequence of hair bundle motion is attributed to the forward transduction itself. Flagella motion in mosquitoes is due to a mechanism separate from sensory transduction. Forward (sensory) transduction can occur when flagella motion (SOs) is blocked (Warren et al., 2010). Is there more than one source of flagella motion in mosquitoes?

We thank the reviewer for this question that touches on the above discussed relation between SOs and transduction, which is as yet unresolved. It cannot be stated at present that transduction and SOs are independent and there is no evidence that suggests that SOs can persist if transduction is blocked. A specific block of only SOs without affecting transduction has not been shown either. The hereto required experiments would involve parallel mechanical and electrophysiological techniques combined with either specific genetic or pharmacological manipulations. This, we are afraid, will have to be left for future experiments.

In analogy to *Drosophila*, it is plausible to assume that dynein based motors underlie the mosquito SOs (as also suggested in Warren et al. 2010, "The dynein-tubulin motor powers active oscillations and amplification in the hearing organ of the mosquito"). Dynein based motors are, however, also contributing to transduction and adaptation (Karak et al., "Diverse Roles of Axonemal Dyneins in *Drosophila* Auditory Neuron Function and Mechanical Amplification in Hearing", Sci Rep-Uk. 2015). We are therefore not in a position to provide a decisive answer to the question if there is more than one source of flagella motion, though this of course remains possible.

Page 24, 17-20. This appears a too simplistic approach (see also comments above). The forces generated by the radially arranged scolopodia will be vectorial. Scolopodia on other prongs will contribute according to the cosine of the angle of the prongs to the flagella displacement. Contributions will also be influenced by the overall geometry of the JO.

We thank the reviewer one more time for this important point, which we have addressed more directly now in the manuscript and also in our answer above.

Page 24 28-30. Couldn't Myosin VII A or Dynein be possible candidates?

This question directly touches on the second holy grail of mechanotransduction research, the very nature of the motors and amplifiers supporting transduction, motors providing adaptation or generating SOs. To all likelihood these are ciliary proteins in the mosquito JO; myosin expression is not very abundant in cilia but we cannot of course exclude a role of myosins. Dyneins, to the contrary, are abundant and have been shown to play important roles.

These ideas have been suggested before. The notion of the mosquito Prestin was made to add a novel candidate to the mix.

We have rephrased the respective sentence:

“This may imply differences in underlying amplificatory mechanisms, potentially involving the two identified mosquito orthologues of the mammalian outer hair cell motor protein Prestin⁴⁶, though myosins and dyneins could also be possible candidates. Although the *Drosophila* Prestin orthologue does not seem to contribute to mechanical feedback amplification⁴⁷, this question still awaits experimental clarification in mosquitoes.”

Page 24, 33-35. Sex and species specific differences could be attributed to differences between species and sexes in JO geometry and relative contribution of sensilla to flagella motion (see also above).

Again, we agree that some of the observed differences may reflect geometric differences. We have amended the manuscript to give credit to this possibility and thank the reviewer again for pointing us into this direction (see above).

P24L33: Is it possible to advance any behavioural and ecological significance for the “substantial sex and species specific differences”?

We have, also in responses to the other two reviewers, tried to make a dedicated effort to suggest some ecological roles for the observed biophysical differences.

Page 25. 2-3. This sentence is not clear. In vertebrates, mechanical feed back is provided by hair bundle (MET) motility (product of forward transduction) and voltage-dependent somatic motility (independent of but dependent on forward transduction). Basis of motility in JO (very likely dynein and ??, coupled to forward transduction?).

We admittedly struggle a little bit to define the terms forward and reverse transduction in any operationally meaningful way for our experiments or thinking, as all force transmission in this system will be inherently reciprocal and also feedback on one another. The sentence describes the commonalities identified between auditory cells in fruit flies, mosquitoes and vertebrates (mostly non-mammalian). Somatic electromotility, which we tried to hint at with our suggestion of a role for mosquito Prestin, is the privilege of mammalian OHCs, but hair bundle based processes can be found across the vertebrate clade. In order to minimise the spatial extent of our paper, and with the permission of the reviewer, we would like to leave this sentence as is.

Page 25, 7-8. There does not appear to be, anywhere in the paper, a justification for making this assumption, apart from others have used as stated here. As already questioned above, it is not apparent how the model copes with such complexities such as opposing gating springs and the geometry of the JO.

We apologise again for not having explained the model we used, and its adequacy, more clearly. We have here relied on the previously published derivations and justifications (most explicitly done in the supplement to Albert et al. 2007, Curr Biol). The opposing MET modules (gating springs, etc.) do not at all impair our approach, but they are in line with our compliance model.

Page 25, 14-21. As already asked above. Could these differences also be accounted, for example, by differences in geometry of the JO and proportions in scolopedial competitive interaction?

Again, we think that some aspects might be caused by, or correlate with, a different transduction geometry.

P25L18: “The males of all species had transducer modules with (i) a greater total gating spring stiffness, KGS, (ii) larger single channel gating forces, z , and (iii) smaller numbers of

predicted transducer channels, N, than conspecific females” – The sex-specific relation between the calculated transducer parameters and the hearing function is an important point of discussion that is lacking; more to the point, how do these parameter values (either taken individually or together) are associated to the higher hearing sensitivity in males?

This is an important point, namely to relate the biophysical data to ecological roles or consequences. We have revised parts of the paper to give more credit to this and also refer to the paper by Hudspeth et al. 2000, “Putting ion channels to work: Mechano-electrical transduction, adaptation, and amplification by hair cells”, PNAS, which gives a good account of how individual parameters can affect system function.

Page 26, 30-32. Similar behaviour has been observed in the mammalian auditory system, especially in CF bats (J. Neurosci., 2003,23:9508 –9518. See Strogatz SH (1994) Nonlinear dynamics and chaos: with applications to physics, biology, chemistry, and engineering. Reading MA: Addison-Wesley. for a theoretical basis of the observations you report.

These are great suggestions, indeed, which will be of great value for future syntheses of auditory function and acoustic communication in mosquitoes. With the reviewer’s permission we would prefer to use, and discuss, these references - which do not directly relate to the data presented in our here submitted original research article - in a follow-up review, in which we would like to hypothesize and synthesize some of the body of theoretical and experimental evidence around insect auditory systems. Thanks for pointing us towards these publications!

Page 27, 7-9. DPs, due to interaction between an external tone and SOs have already been reported (Current Biology 20, 131–136). Difference tones (quadratic) DPs, (the product of a stiffening nonlinearity like that reported in your MS) are recorded near the apical surface of the JO. cubic distortion, likely to be generated at the level of the neurons, is recorded near the base of the JO (Lapshin, D. N. (2012) Entomol Rev, 92, No. 6, pp. 605–621). Of considerable importance, as mentioned above, is the level of an external tone (female flight tone) required to entrain the SOs.

As we here conduct and report compound recordings, we cannot speculate on the local origin of specific DPs. They could all be recorded - in filtered forms - in the flagellar motion and the antennal nerve. The level requirements for entrainment are indeed an important next question, as we explained earlier. We fully agree that this will have to be one of the immediate next steps in the experimental progression, as we explicitly mention this in our manuscript, and have cited both papers included in this comment.

Minor points

P2L22: “Several studies have proposed potential mechanisms of acoustic signalling between conspecific males and females within these swarms.” – Please reference these studies.

Thanks for highlighting this. We have modified the sentence and added the requested references.

The sentence now reads:

“Several studies have proposed potential mechanisms of acoustic signalling between conspecific males and females^{10, 11, 12, 13, 17, 18} but few have discussed these within the context of flying animals^{19, 20} or related these to the specific environment of the swarm¹⁹.”

P3L1: “Another phenomenon that might offer valuable insights into mosquito acoustic courtship are spontaneously occurring ... SOs...” – I would suggest that the SOs might first offer insights into the sensory function, well before the mating behaviour. But consider this just a suggestion.

Thanks for highlighting this. We have changed the respective sentence.

The sentence now reads:

“Another phenomenon that might offer valuable insights into mosquito auditory function (and indeed acoustic courtship) are spontaneously occurring, *self-sustained oscillations* (SOs) of the flagellum.”

P3L5: Are any references that support that the SOs, and in mosquitoes in particular, are a “pathological signature”?

No such reference exists, though neither does any reference that shows any physiological function of SOs. At present this is the question that is being asked most regularly when presenting, or discussing, our data at meetings.

P3L16: “METs” – This abbreviation wasn’t used anywhere in the text.

Thanks for highlighting this. We have removed the abbreviation from the text.

P4L1 and throughout – The authors favour the term the “passive” in the text and table1 but use the term “sedated” in figure 1 (and legend). I would suggest using the same term in all instances.

Thanks for highlighting this. We have changed the respective parts.

P4L8 and tables: “SE” – Are these values the Standard Errors of the median?

Yes, SE is for standard error

P7L9: “Calculated energy gains” – Shouldn’t be power gains?

Thanks for highlighting this. We have changed the respective parts.

P9L25: “Female *An. gambiae* produced CAP responses of significantly smaller magnitudes than those seen in females from the two other species” – A statistical test is needed to support this.

Thanks for highlighting this. We have now modified this and included the table of values for the statistical test in the supplement.

P12L11: “Bi-dimensional error bars utilise standard errors.” – What error bars?

Thanks for highlighting this. The bars are very small but present.

P13: Table 3 – Please define in the legend all the other parameters.

Thanks for highlighting this. We have changed the respective parts.

P14L6: “Paying tribute to their potential roles in modulating mosquito auditory sensitivity, specific focus was placed on the JOs’ efferent innervation patterns.” – Awkward sentence.

Thanks for highlighting this. We have changed the respective parts.

The sentence now reads:

“Specific focus was placed on the JOs’ efferent innervation patterns due to their potential roles in modulating auditory sensitivity.”

Fig. 4A and legend: The Left/right separation of data in fig 4A is confusing. Why not separate them in 2 different figures?

Thanks for highlighting this. We have tried to clarify the figure by a clearer labelling and by following the suggestions from the other reviewers.

Figure 5A: It would be useful to identify the different rows with the corresponding quantification: PT Stimulus, flagellar displacement, AN responses, Power spectra.

Thanks for highlighting this. We have changed the respective parts.

P25L31: “Conversion of the $\square 90$ displacements into angular deflections...” – Since angular deflections facilitates comparisons with other studies, it would be useful to include these values in Table 2.

Thanks for highlighting this. Please refer to table 2 for these angular lambda 90 values.

- I haven’t found any reference in the manuscript on if and how the erection of the antennal hairs in *A. gambiae* males was achieved and/or controlled.

Thanks for highlighting this. We have added a section to the Methods. The erection state was monitored via a video camera that is co-axially aligned with our LDV setup. Fibrillae were not erect during our experiments. However, erection status for electrostatic actuation would only (if anything) affect the transfer function between voltage and force, which is not relevant for any analyses conducted (as we plot and analyse all data against the final effective force). We measure the antennal displacement and we measure the force, the voltage to produce these is only of technical interest. This is in stark contrast to acoustic stimulation where the erection state will directly affect the acoustic sensitivity.

The sentence reads:

“Whilst male *Ae. aegypti* and *Cx. quinquefasciatus* antennal fibrillae are permanently erect, those of male *An. gambiae* are erect only during strict circadian time windows associated with swarming behaviour [REF]. All recordings were made within a 2 hour time window beginning 1 hour after light onset – thus male *An. gambiae* fibrillae were not erect throughout these experiments.”

Additional changes not directly requested by the reviewers:

We added the highlighted sentence to the below paragraph in the discussion:

Taken together such an auditory system would enable the male to detect, and amplify, a faint female flight tone by ‘locking into’ the female wing beat frequency and using low-frequency DPs of the amplified female flight tone and his own wing beat frequency. As reported before^{12, 60}, the nerves of all males tested here were most sensitive to stimulus frequencies around

these predicted low-frequency DPs (data not shown but available). By using DPs rather than the original flight tones, males could turn the 'noise' of their own wingbeat into a signal amplifier (Figure 5C). The ears of male mosquitoes would thus form a biological equivalent of a *superheterodyne receiver*, or *superhet*; virtually all modern radios operate according to the *superhet* principle [REF Armstrong EH. A new system of short wave amplification. *Proceedings of the Institute of Radio Engineers*. 1921;9(1):3-11.]. Future studies will have to test this proposal for naturally occurring levels of male and female wing beats.

REVIEWERS' COMMENTS:

Reviewer #1 (Remarks to the Author):

The revised manuscript was an absolute pleasure to read. The amount of work that is contained within is staggering, important, and well explained. The authors present the most careful dissection of mosquito hearing to date. The findings are novel and will influence thinking in the fields of mosquito sensory ecology and control.

I have some small suggestions for the discussion, but otherwise I am very happy to recommend publication and look forward to seeing it out.

On page 27, Line 16-17: You mention that females may use sound in host seeking. There are many papers that have tried to incorporate sound lures (Reviewed in Service 1993) in general, but those that look at the role of sound in host-seeking specifically have focused on frog calls. Frog calls have been found attractive for both *Culex* and *Uranotaenia* (Borkent and Belton 2006, Toma et al. 2005) mosquitoes. It might be worth clarifying this.

On page 28 Line 11-12. I recommend rephrasing this sentence for clarity to read: " Intriguingly, our data shows that one of the main differences between male and female ears is the gating properties of their auditory transducers:.."

Reviewer #2 (Remarks to the Author):

The authors have addressed my concerns in the revised manuscript. I have no further comments or suggestions .

Reviewer #3 (Remarks to the Author):

We thank the author for the full responses to our comments. Changes made to the paper fully address the issues we raised.

RESPONSE TO REVIEWERS' COMMENTS:

Reviewer #1 (Remarks to the Author):

The revised manuscript was an absolute pleasure to read. The amount of work that is contained within is staggering, important, and well explained. The authors present the most careful dissection of mosquito hearing to date. The findings are novel and will influence thinking in the fields of mosquito sensory ecology and control.

I have some small suggestions for the discussion, but otherwise I am very happy to recommend publication and look forward to seeing it out.

On page 27, Line 16-17: You mention that females may use sound in host seeking. There are many papers that have tried to incorporate sound lures (Reviewed in Service 1993) in general, but those that looking at the role of sound in host-seeking specifically have focused on frog calls. Frog calls have been found attractive for both Culex and Uranotaenia (Borkent and Belton 2006, Toma et al. 2005) mosquitoes. It might be worth clarifying this.

We thank the reviewer for the comment and have now clarified this point. The section now reads:

Also, roles of audition beyond mating might include predator avoidance (in both males and females)⁴³ or host finding (in females). Corresponding phonotactic responses related to frog calls have indeed been reported for females of frog-biting mosquito species^{44, 45}, including *Culex* spp.⁴⁶. This possibly explains why *Cx. quinquefasciatus* was the only species in our study where female baseline auditory amplification exceeded that of males.

On page 28 Line 11-12. I recommend rephrasing this sentence for clarity to read: "Intriguingly, our data shows that one of the main differences between male and female ears is the gating properties of their auditory transducers.."

This sentence has been changed accordingly. It now reads:

Intriguingly, our data shows that one of the main differences between male and female ears is the gating properties of their auditory transducers

Reviewer #2 (Remarks to the Author):

The authors have addressed my concerns in the revised manuscript. I have no further comments or suggestions.

Reviewer #3 (Remarks to the Author):

We thank the author for the full responses to our comments. Changes made to the paper fully address the issues we raised.